# Intranasally administrated fusion-inhibitory lipopeptides block SARS-CoV-2 infection in mice and enable long-term protective immunity
Said Mougari [1,15], Valérie Favède [1,2,15], Camilla Predella [3,4,15], Olivier Reynard [1], Stephanie Durand[1], Magalie Mazelier [1], Edoardo Pizzioli [1], Didier Decimo [1], Francesca T. Bovier[3], Lauren M. Lapsley[3], Candace Castagna [5], Nicole A. P. Lieberman [6], Guillaume Noel[7], Cyrille Mathieu [1], Bernard Malissen[8], Thomas Briese [9], Alexander L. Greninger [6], Christopher A. Alabi [10], N. Valerio Dorrello[3], Stéphane Marot [11], Anne-Geneviève Marcelin [11], Ana Zarubica[8], Anne Moscona [3,12,13], Matteo Porotto [3,12,14] ✉ & Branka Horvat [1] ✉

We have assessed antiviral activity and induction of protective immunity of fusion-inhibitory lipopeptides derived from the C-terminal heptad-repeat domain of SARS-CoV-2 spike glycoprotein in transgenic mice expressing human ACE2 (K18-hACE2). The lipopeptides block SARS-CoV-2 infection in cell lines and lung-derived organotypic cultures. Intranasal administration in mice allows the maintenance of homeostatic transcriptomic immune profile in lungs, prevents body-weight loss, decreases viral load and shedding, and protects mice from death caused by SARS-CoV-2 variants. Prolonged administration of high-dose lipopeptides has neither adverse effects nor impairs peptide efficacy in subsequent SARS-CoV-2 challenges. The peptide-protected mice develop cross-reactive neutralizing antibodies against both SARS-CoV-2 used for the initial infection and recently circulating variants, and are completely protected from a second lethal infection, suggesting that they developed SARS-CoV-2-specific immunity. This strategy provides an additional antiviral approach in the global effort against COVID-19 and may contribute to development of rapid responses against emerging pathogenic viruses.

Severe acute respiratory syndrome coronavirus 2 (SARS-CoV-2) infection requires fusion of the viral and host cell membranes, which occurs either at the target cell surface or at the endosomal membrane. This critical step of viral entry is mediated by the viral spike glycoprotein (S), which is an important target for the development of both preventive and therapeutic antiviral approaches[1,2]. Similar to other coronaviruses, the S of SARS-CoV-2 is a type-I fusion protein, assembled as a multiple copy homotrimer anchored to the outer virion membrane by a transmembrane (TM) domain[3–5]. The protein is cleaved at the surface of mature virions into two non-covalently associated subunits, S1 and S2 (Fig. 1a), in a process thought to be catalyzed by furin-like proprotein convertases during virion morphogenesis in virus-infected cells[3,4]. During viral entry, S, like other class I viral fusion proteins, folds into an energetically stable state to overcome the repulsive charges that arise between the virus and host cell membranes

during membrane fusion[6]. After receptor binding mediated by the S1 sub-unit, S undergoes a cascade of conformational changes triggered by host proteases such as TMPRSS2 and cathepsin-L that cleave S at its second cleavage site located at the S2′ site in the S2 subunit[7–9]. Fusion starts with the emergence of the hydrophobic fusion peptide (FP) of S2 from the head group and its extension and insertion into the host cell membrane. Subsequent zippering of two heptad repeat domains located in the S2, one near the amino N terminus (HRN) and the other near the C terminus (HRC), into a stable hexa-helical bundle conformation, provides energy for virion and host cell membranes merger[10–12].

Peptides derived from viral fusion proteins are potent entry inhibitors. One example is enfuvirtide (T-20 or Fuzeon), a Food and Drug Administration-approved synthetic 36-amino acid synthetic peptide targeting the HIV type 1 (HIV-1) envelope glycoprotein 4[13]. A similar strategy

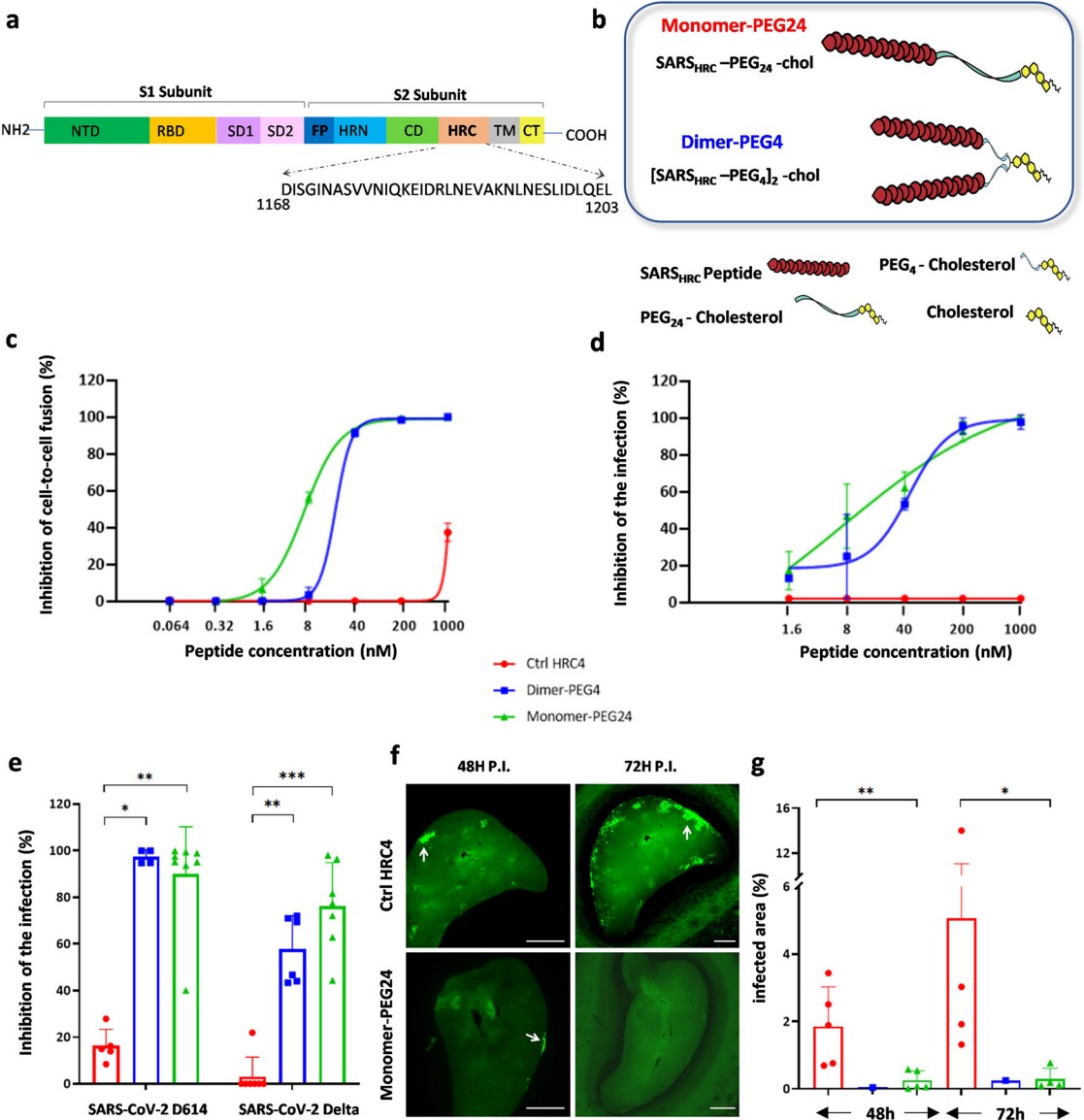

**Fig. 1 | Fusion inhibitory lipopeptides block SARS-CoV-2-Spike-mediated fusion in hACE2-transfected cells and SARS-CoV-2 replication in vitro and ex vivo. a** Schematic of SARS-CoV-2 Spike protein primary structure with peptide sequence. NTD N-terminal domain, RBD receptor-binding domain, SD1 subdomain-1, SD2 subdomain-2, FP fusion protein, HRN heptad repeat N-terminal, CD connector domain, HRC heptad repeat C-terminal, TM transmembrane domain, CT cytoplasmic tail. **b** Schematic presentation of the composition of the monomer-PEG24 (SARS$_{HRC}$-PEG$_{24}$-chol) and dimer-PEG4 ([SARS$_{HRC}$-PEG$_4$]$_2$-chol) peptides. **c** Cell–cell fusion inhibition assay in 293T-ACE2 cells with 293T-Spike Wuhan D614. Data are means ± standard deviation (SD) (error bars) from three separate experiments with the curve representing a four-parameter dose–response model (Ctrl HRC4: control peptide, measles-specific). **d** Virus inhibition assay, measuring peptide-mediated inhibition of SARS-CoV-2 (Wuhan D614) infection (100 plaque-forming unit- PFU/well) in Vero cells. Percent inhibition was calculated as the ratio of PFU in the presence of a specific concentration of inhibitor and the PFU in the absence of inhibitor. Data are means ± SD from three separate experiments. **e** Inhibition of infection in organotypic cultures from lungs of K18-hACE2 mice with either recombinant SARS-CoV-2 neon-green or SARS-CoV-2 Delta (500 PFUs), with peptides added into the culture during 96 h, assessed using RT-qPCR (4–8 slices per condition, generated from 3 mice, results presented as mean ± SD). **f** Spread of fluorescent rSARS-CoV-2 neon-green virus in representative fluorescence micrographs of lung organotypic cultures, in the presence and absence of peptide (arrows indicate the sites of viral replication in cultures, (scale bars: 1 mm). **g** Quantification of the percentage of infected area, detected by immunofluorescence on scanned slides using QuPath, Open-source software for digital pathology image analysis, results presented as mean ± SD (*$p < 0.05$, **$p < 0.01$, ***$p < 0.001$, Mann–Whitney test).

has been applied to target coronaviruses, including the original severe acute respiratory syndrome coronavirus (SARS-CoV)[14,15], Middle East respiratory syndrome coronavirus[16,17], and the more recent SARS-CoV-2[18–22]. Peptides derived from the HRC domain of the S have been shown to inhibit the refolding of S by targeting the extended transient form of the S trimer and blocking its conformational changes. These peptides are promising candidate compounds to block S-mediated fusion, entry of SARS-CoV-2 into host cells, and infection[12,20,21].

To further improve the efficacy of HRC-based inhibitors, various structural engineering approaches have been used. A human coronavirus OC43 HRC-derived peptide, EK1, was mutated to enhance its stability, solubility, and antiviral activity[23], resulting in a half-maximal inhibitory concentration (IC$_{50}$) of ~300 nM in a SARS-CoV-2 S-based cell-to-cell fusion assay[18] and prevention of SARS-CoV-2 infection[24]. We conjugated a SARS-CoV-2 HRC-derived peptide to polyethylene glycol and cholesterol to facilitate peptide targeting to plasma membranes, resulting in an IC$_{50}$ of

~5 nM in SARS-CoV-2 infection assays and in ex vivo human airway epithelial cell cultures. These lipid-conjugated fusion inhibitory peptides completely prevented transmission of SARS-CoV-2 in ferrets[20], an excellent model for transmission studies since the ferret develops upper respiratory tract infection[25,26]. In other studies, a SARS-CoV-2-derived HRC peptide was stabilized by the introduction of hydrocarbon staples, but the half-maximal effective concentration levels remained in the micromolar range[27]. While in these approaches a 36 amino-acid long segment of SARS-CoV-2 S (residues 1168–1203) was used, it was recently suggested that N-terminal extension of peptides by 6–7 amino acids confers improved S1-binding and antiviral activity[28,29]. Dimerization of fusion inhibitory peptides may increase their anti-SARS-CoV-2 entry activity[20,30], as we previously showed for peptides designed to inhibit infection by other viruses[31]. In contrast to the rapidly mutating S1 subunit, which contains the virus receptor-binding domain (RBD), the S2 subunit that contains the HRC domain target of the antiviral peptides (Fig. 1a) is highly conserved in newly emerging SARS-CoV-2 variants of concern (VOC)[32,33], allowing HRC-derived peptides to retain efficient inhibitory activity against VOCs[22]. Further modifications of HRC-derived peptides have been proposed, including chemical stapling[34] or coupling to ACE2-derived peptides[35].

More efficacious antivirals are urgently needed to combat COVID-19, with constantly mutating SARS-CoV-2 variants in circulation. The potency of fusion inhibitory lipopeptides, shown previously to prevent transmission of SARS-CoV-2 infection in ferrets[20], has been investigated in this present study in mice. We assessed, in addition, whether the HRC-derived peptides permit the generation of long-term anti-viral immunity in protected mice. We used transgenic mice expressing the human angiotensin I-converting enzyme 2 (ACE2) receptor driven by the cytokeratin-18 (K18) gene promoter (K18-hACE2), shown to be a model that mimics the clinical manifestations and immune disorders of severe COVID-19 infection in humans[36,37]. We show that these lipopeptides protect mice from lethal SARS-CoV-2 infection and permit the establishment of long-lasting immunity against SARS-CoV-2 reinfection in protected animals, as well as the generation of cross-neutralizing antibodies to the recently evolved Omicron XBB variant. Multiple administrations of high-dose lipopeptides did not result in adverse effects and did not interfere with peptides' antiviral protection during subsequent SARS-CoV-2 challenge.

## Results

### Fusion inhibitory peptides inhibit SARS-CoV-2-S mediated fusion and virus infection in vitro and ex vivo

Two peptides described in earlier studies that efficiently inhibit SARS-CoV-2 infection in vitro and transmission of the virus in the ferret model[20-22] were analyzed here for their ability to protect mice against SARS-CoV-2. Both peptides are derived from the HRC domain of the S and contain a 36-amino acid sequence corresponding to residues 1168–1203 (Fig. 1a), conjugated to cholesterol. Conjugation to a lipophilic moiety has been shown to increase peptide potency by extending the in vivo half-life and increasing local concentration at the infection site by anchoring the peptides in the cell membrane[31,38-40], and PEGylation was suggested to enhance muco-penetration, being advantageous for the inhalation of aerosol delivery[41]. The first peptide, SARS$_{HRC}$–PEG$_{24}$–chol, contains a monomeric HRC domain linked to 24 molecules of polyethylene glycol (**monomer-PEG24**), while the second one, [SARS$_{HRC}$–PEG$_4$]$_2$–chol contains a dimeric HRC group bearing four molecules of polyethylene glycol (**dimer-PEG4**) (Fig. 1b). Addition of an extended flexible PEG linker between the C-terminal leucine of the peptide and the cholesterol moiety (PEG-24) was shown to improve solubility and increase the peptide's ability to promote rapid binding and insertion in lipid membranes, likely by limiting local steric hindrance[42].

We first evaluated peptide inhibition of S-mediated cell-cell fusion using a quantitative β-galactosidase (β-Gal) complementation assay (Fig. 1c). A dimeric lipopeptide derived from the HRC domain of the F protein of measles virus (HRC4)[43] was used as a negative control. In contrast to the measles-derived peptide, both SARS-CoV-2 S-derived peptides potently inhibit S-mediated cell–cell fusion with an IC$_{50}$ of 20.27 nM for the

dimeric and 6.89 nM for the monomeric peptide, in line with previously published results[20-22], validating the peptide preparation used in this study. We then assessed the capacity of the lipopeptides to inhibit SARS-CoV-2 infection in Vero cells (Fig. 1d). The S-derived lipopeptides demonstrated a potent inhibitory effect against SARS-CoV-2 entry in Vero cells, in contrast to the control measles HRC4 lipopeptide with IC$_{50}$ comparable to those from the fusion assay (45.64 and 7.08 nM for the dimer-PEG4 and the monomer-PEG24, respectively).

We next analyzed the effect of peptides on SARS-CoV-2 infection in an organotypic lung culture (Fig. 1e–g). Organotypic cultures preserve three-dimensional lung architecture and are composed of different cell populations that are representative of local cell organization, allowing these cultures to mimic different steps of virus infection that occur in vivo[44]. We tested the ability of fusion inhibitory lipopeptides to block SARS-CoV-2 D614 and Delta variant entry and spread in lung organotypic cultures prepared from K18-hACE2 transgenic mice. Cultures were treated with either the monomer-PEG24 or the dimer-PEG4 for 90 min before the infection and then infected with 500 plaque-forming units (PFU) of SARS-CoV-2 either D614, delta variant, or SARS-CoV-2-mNG, for immunofluorescence analysis (Fig. S1). Organ slices were collected at 96 hours post-infection (h.p.i.) for RNA extraction and RT-qPCR targeting the nucleocapsid (N) RNA of SARS-CoV-2, in parallel with fluorescence microscopy to monitor the effect of peptide on virus spread in culture. Both lipopeptides inhibited SARS-CoV-2 D614 and Delta infection in lung organotypic cultures (Fig. 1e). In the absence of the virus-specific peptide treatment, virus replication occurred at the periphery of infected lungs at 48 h.p.i., expanding across the organ at 72 h.p.i., (Fig. 1f). Analysis of scanned images using digital pathology imaging software showed significant inhibition of virus propagation in cultures treated with lipopeptide at 48 and 72 h.p.i. (Fig. 1f, g). These data support the efficacy of fusion inhibitory peptides in blocking the spread of SARS-CoV-2 in mouse lungs cultured ex vivo. As our previous studies showed a correlation between treatment efficacy in ex vivo models and efficacy in animal models for other viruses, including measles virus[40], these data supported the potential of fusion inhibitory peptides for study in vivo. In addition, we analyzed the possible toxicity of repeated administration of lipopeptides in mouse olfactory bulb, cerebellum, and lung organotypic cultures (Fig. S2). We did not observe any effect on the metabolic activity of these cultures after 3-day peptide treatment, reinforcing the safety profile of lipopeptides for further in vivo application.

### Pre-treatment with fusion inhibitory peptides protects K18-hACE2 mice from SARS-CoV-2 infection

We analyzed the biodistribution of lipopeptides in mice by measuring peptide concentration in mouse lungs at several time points: 8, 24, and 48 h after intranasal (i.n.) dosing (Fig. S3). In agreement with our previous study[20], both lipopeptides were found at high lung concentrations at 24 h, and monomer-PEG24 remained at a high concentration at 48 h. We thus proceeded to an in vivo study to assess both fusion inhibitory lipopeptides against SARS-CoV-2 infection in K18-hACE2 transgenic mice, using i.n. peptide administration in 24 h intervals. The infectious dose of SARS-CoV-2 D614 was chosen based on the initial experiments, aiming to obtain 100% lethality in K18-hACE2 mice. Omicron was excluded from further in vivo studies due to the low mortality and viral load observed in infected mice (Fig. S4).

K18-hACE2 mice were randomized into three groups. Two groups were pretreated with either the monomer-PEG24 or the dimer-PEG4 peptide, following the protocol shown in Fig. 2a, while the control group received only the vehicle used to resuspend the peptide solution. All mice were infected i.n. with SARS-CoV-2 D614 (10$^4$ PFU in 30 μL) and weighed daily. While the mock-treated group experienced significant body weight loss starting at 1 dpi and reaching a maximum of ≥25% of body weight loss by 8 dpi, the weight in the lipopeptide-treated groups did not show significant variation (Fig. 2b). All mock-treated mice succumbed to SARS-CoV-2 infection by 9 dpi (Fig. 2c). In contrast, while only one dimeric peptide-treated mouse died from the infection by 10 dpi, all monomer-

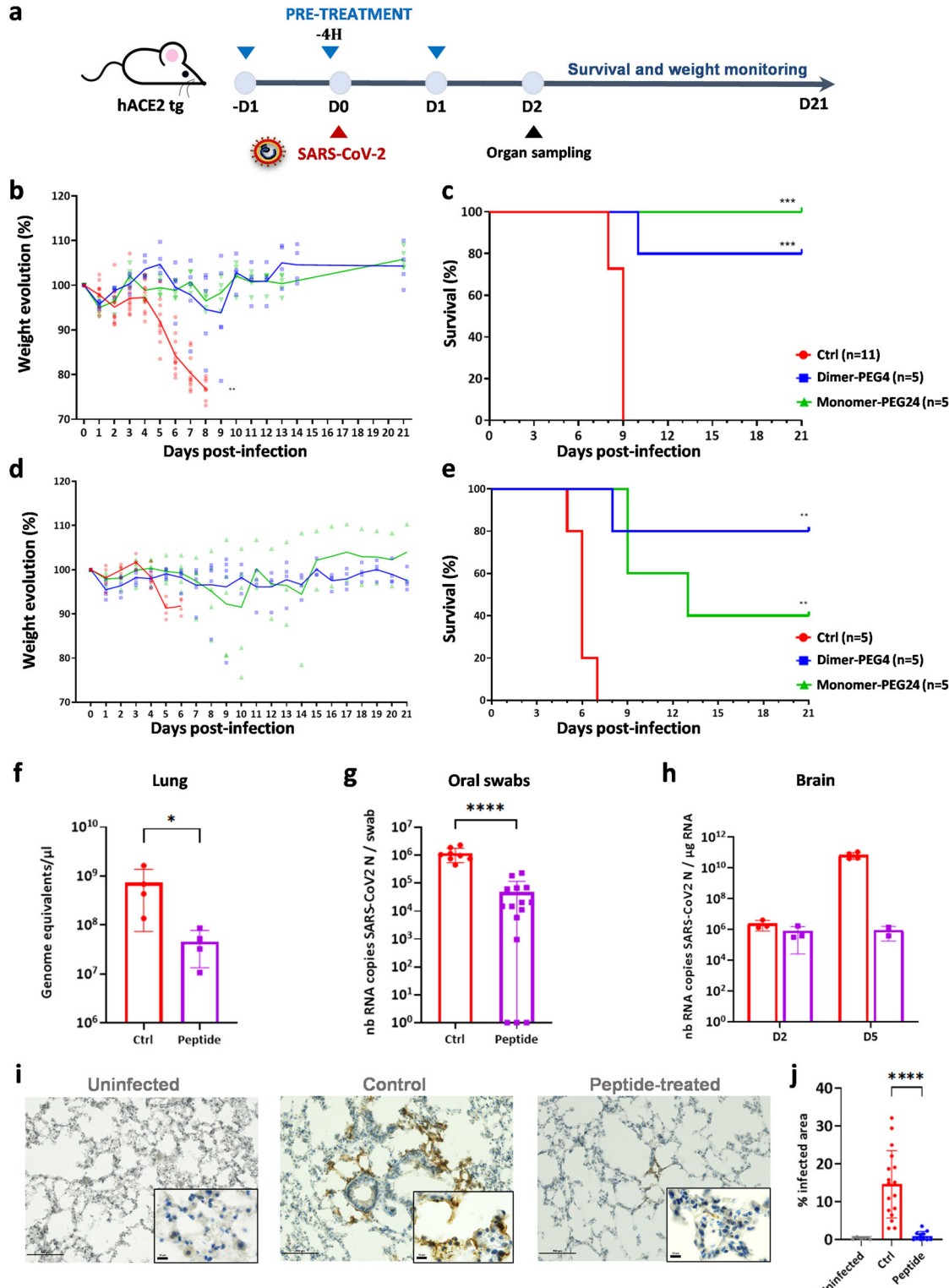

PEG-24-treated mice survived SARS-CoV-2 infection (100% survival), and completely recovered their initial weight by 10 dpi.

To further analyze the effect of lipopeptides in the infections with SARS-CoV-2 VOCs, we evaluated the effect on infection with the SARS-CoV-2 Delta variant. In line with previously published data[43,44], the Delta variant required a higher lethal dose than the wild-type SARS-CoV-2 in K18-hACE2 mice and was more pathogenic than the Omicron variant (Fig. S4). We used the dose of Delta variant, shown to induce 100% mortality

($10^5$ PFU in 30 μL), applying the same experimental protocol shown in Fig. 2a. Similar to D614, all the untreated mice succumbed to the infection whereas pretreatment with lipopeptides protected mice from weight loss (Fig. 2d) and significantly increased their survival (Fig. 2e), showing the protective effect of lipopeptides against both SARS-CoV-2 variants in vivo. Enhanced pathology and inflammatory responses induced by the SARS-CoV-2 Delta variant in K18-hACE2 mice[45] may be responsible for the slightly lower efficacy of peptides against this variant. Although both

**Fig. 2 | Pretreatment with fusion inhibitory peptides protects K18-hACE2 mice from SARS-CoV-2-induced pathology. a** Experimental design: K18-hACE2 mice received either lipopeptide or vehicle (2% DMSO) intranasally daily for 3 days and were infected with SARS-CoV-2 intranasally 4 h after the second peptide administration or after receiving a vehicle solution (control), and followed for 21 days. **b–e** Body weight normalized to their initial weight on the day of infection and survival of mice following the infection with SARS-CoV-2 D614 ($10^4$ PFU) (**b** and **c**), or Delta variant ($10^5$ PFU) (**d** and **e**). Statistical significance of the effect of peptides on the evolution of weight was determined using a Mann-Whitney test and survival was measured using a Mantel–Cox test, **$p < 0.01$, ***$p < 0.001$. **f** Viral load in lungs (4 mice/group) determined by RT-qPCR, 2 dpi) with SARS-CoV-2 D614. **g** Viral load in oral swabs of mice (8 control mice and 15 peptide-treated mice) at 2 days post-infection (**h**) and in brains (7 control mice and 5 peptide-treated mice) at the indicated time points, using SARS-CoV-2 Delta, determined by RT-qPCR. Results obtained from two lipopeptide-treated series were grouped, results presented as mean ± SD, and statistical significance was determined using a Mann–Whitney test (*$p < 0.05$, ****$p < 0.0001$). **i** Immunohistological analysis of lung sections of mice (2 and 3 animals/condition) that received the indicated treatments 2 dpi with SARS-CoV-2 D614. Staining with anti-SARS-CoV-2 N-specific Ab is indicated with red arrows (scale bars: 500 μM for lower magnification and 10 μM for higher magnification). **j** Quantification of the percentage of infected cells, detected on scanned slides (infected controls, $n = 17$; uninfected, $n = 5$ and peptide-treated infected, $n = 11$), using QuPath, Open-source software for digital pathology image analysis (****$p < 0.0001$, Mann–Whitney test).

monomer-PEG24 and dimer-PEG4 showed significant protective effects compared to the untreated group, the difference between them was not statistically significant.

We next analyzed whether the beneficial effect of lipopeptides on the survival of mice is associated with inhibition of SARS-CoV-2 replication in lungs, brain, and buccal mucosa by assessing the level of viral transcription in mice at 2 dpi (Fig. 2f–h). As both peptides were similarly protective in mice, they were analyzed together and compared to the untreated group. SARS-CoV-2 transcription in lungs (Fig. 2f) as well as in oral mucosa (Fig. 2g) of peptide-treated mice were significantly decreased compared to the control. In addition, lipopeptide pre-treatment reduced viral load in brains when measured at 5 dpi (Fig. 2h). Finally, lung sections from lipopeptide and mock-treated mice were collected at 2 dpi and stained with anti-SARS-CoV-2 N antibody. While SARS-CoV-2 N staining was widespread in the lungs of the mock-treated group, as shown in the representative images in Fig. 2i, the presence of SARS-CoV-2 antigens in lipopeptide-treated groups was significantly decreased, measured by the quantification of infected cells on scanned images (Fig. 2j).

### Transcriptomic profiles in lungs of SARS-CoV-2-infected K18-hACE2 mice pretreated intranasally with fusion inhibitory lipopeptides

To determine whether lipopeptide antiviral treatment affects inflammatory response, we performed RNA-seq analysis on RNA extracted from the lungs of K18-hACE2 mice pretreated intranasally with fusion inhibitory lipopeptides and compared the results with vehicle-treated infected and uninfected mice (Fig. 3) Results are presented as gene expression heatmaps for individual animals (Fig. 3a). The adjusted $p$-value < 0.001 was defined as significant, resulting in 1474 genes significantly modulated in the infected ($n = 3$) compared to uninfected ($n = 3$) groups, and 67 genes significantly modulated in the dimeric peptide-pre-treated group ($n = 2$) vs. the uninfected group. Heatmaps depicting the average $\log_2$ fold change of the 65 differentially expressed genes with the lowest adjusted $p$ values in the lungs of K18-hACE2 mice infected with SARS-CoV-2 D614, were compared to noninfected mice and to mice pretreated with either dimer-PEG4 or monomer-PEG24 lipopeptide, 2 dpi, both shown to have a similar protective effect in K18-hACE2 mice in Fig. 2. Differentially expressed genes with the lowest adjusted $p$ values were assigned into three groups using their reactome[46] and GO BioProcess annotations[47] and the clustering: Interferon/Innate-enriched, Adaptive-enriched and Non-immune genes. In agreement with what has been reported previously[48], CXCL10 (interferon gamma-induced protein 10, IP-10) was increased in infected lungs. Based on the level of expression changes per sample and its very high fold-change increase, CXCL10 did not fit into the Innate or Adaptive-enriched clusters and had to be assigned to the separate category "Immune Other".

Figure 3b presents these results using adaptive/innate/interferon metadata bars based on the non-redundant lists of genes from Reactome/GO[46]. The sum of transcripts per million (TPM) of each gene category from 3A per sample was determined, and the results are presented for each of the three analyzed clusters by box-plots showing individual values and the average sum TPM for each group. Comparison between groups showed that SARS-CoV-2 infection significantly upregulates the expression of genes belonging to innate/interferon and adaptive immune clusters and decreases the expression of non-immune gene clusters, while pretreatment with lipopeptides alleviates the effect of SARS-CoV-2 infection, allowing for the maintenance of a gene expression pattern similar to what is seen in lungs of uninfected animals. These results were further confirmed by RT-qPCR analysis in lung RNA samples for three selected up-regulated transcripts in infected vs. untreated groups and corresponding to Myxovirus Resistance-1 (MX1), Interferon-induced protein with tetratricopeptide repeats 1 (IFIT1), and CXCL10 and for one transcript which did not show differential expression: CXCL11 (C-X-C motif chemokine 11) (Fig. 3c). Taken together, these results suggest that treatment with fusion-inhibitory lipopeptides reduces the lung inflammatory response to SARS-CoV-2 infection in mice, likely the result of reduced viral infection.

### Effect of multiple administrations of high-dose lipopeptides in mice and lack of interference with antiviral protection

The anti-fusion lipopeptides offer an antiviral prophylactic approach that may need to be administered frequently. We thus assessed whether multiple administrations of high-dose lipopeptides have any adverse effects in mice or interfere with peptide-induced antiviral protection during subsequent SARS-CoV-2 challenges. As the SARS-CoV-2 variant Alpha had emerged in the UK with higher transmissibility and pathogenicity than previously circulating strains[49], we used this variant for further analysis. The infectious dose of SARS-CoV-2 Alpha-UK was chosen based on the initial experiments, aiming to obtain 100% lethality in K18-hACE2 (Fig. S4b). We first analyzed whether repeated i.n. administrations of high-dose lipopeptide over four weeks (four treatments/week with a dose of 20 mg/kg) induced peptide-specific autoantibodies in 24 K18-hACE2 mice (Fig. 4a). Animals were tested two weeks after receiving the last dose of dimer-PEG4 and the presence of peptide-specific antibodies was detected by ELISA in 50% of peptide-treated animals (Fig. 4b). These mice were then separated into two groups, with half of each group positive for peptide-specific antibodies. One group received an additional peptide treatment (group 1), and the other remained without treatment (group 2). Two groups of naïve K18-ACE2 mice (12 mice in each group) were included in the experiment: one received peptide treatment (group 3) and the other only vehicle (group 4).

All mice were infected i.n. with SARS-CoV-2 Alpha-UK ($10^4$ PFU/mouse, in 40 μL) on day 0 and treated with peptide i.n. each day for five days (days −1, 0, 1, 2, and 3, at 20 mg/kg of peptide) and followed for weight loss (Fig. 4c) and survival (Fig. 4d). Groups receiving only the initial peptide pretreatment (group 2) or receiving no treatment (group 4) lost weight starting from day 4 pi, and all succumbed to the infection by day 9 pi. In contrast, all mice in the two other groups—either those pretreated with a high dose of peptide (with 50% of mice having peptide autoantibodies in the serum) and treated for five days (group 1), or those receiving only the second peptide treatment (group 3)—survived lethal-dose SARS-CoV-2 infection (Fig. 4c, d). Mice were monitored daily, and all surviving mice were sacrificed at 10 dpi. Organs (brain, lungs, liver, kidney, and intestines) were extracted, and organ-specific viral load was analyzed by RT-qPCR (Fig. 4e). Significant differences between groups surviving or succumbing to infection were found in brains, lungs, livers, and kidneys.

**Fig. 3 | Intranasal administration of fusion inhibitory lipopeptides in SARS-CoV-2-infected mice modifies transcriptomic profile in lungs at 2 days post-infection. a** Heatmaps depicting the average log$_2$ fold change of the 65 differentially expressed genes with the lowest adjusted *p* values in lungs of K18-hACE2 mice infected with SARS-CoV-2 D614 (*n* = 3), compared to noninfected mice (*n* = 3) and to mice pretreated with either dimer-PEG4 or monomer-PEG24 lipopeptides (*n* = 3), 2 days post-infection. Red denotes up-regulated and blue denotes down-regulated transcripts. **b** Differentially expressed genes with the lowest adjusted *p* values were placed into three groups, using their annotations and the clustering: Interferon/Innate-enriched, Adaptive-enriched, and Non-immune genes. Results are presented with adaptive/innate/interferon metadata bars, based on the non-redundant lists of genes from reactome+GO BioProcess. The sum of transcripts per million (TPM) of each gene category per sample was determined, and results are presented for each of the 3 clusters by box-plots showing individual values and the average sum TPM for each group. Comparisons between uninfected, infected-untreated, and infected-peptide-treated groups were performed using ANOVAs with Tukey's Post Hoc test (***p* < 0.01). **c** RT-qPCR analysis in analyzed lung samples from 3 mice/group of selected three up-regulated transcripts in the infected-untreated group, MX1, IFIT1, and CXCL10, and one transcript where differential expression was not observed (CXCL11). Results are expressed as fold change relative to the number of copies of mRNA in each group, compared to the uninfected group. Uninf: Uninfected, untreated group, Infected: Infected, untreated group; Peptide: infected, pretreated with either dimer or PEG-24 monomer group; results presented as mean ± SD (**p* < 0.05, ***p* < 0.01, ****p* < 0.001, *****p* < 0.0001 Mann–Whitney test).

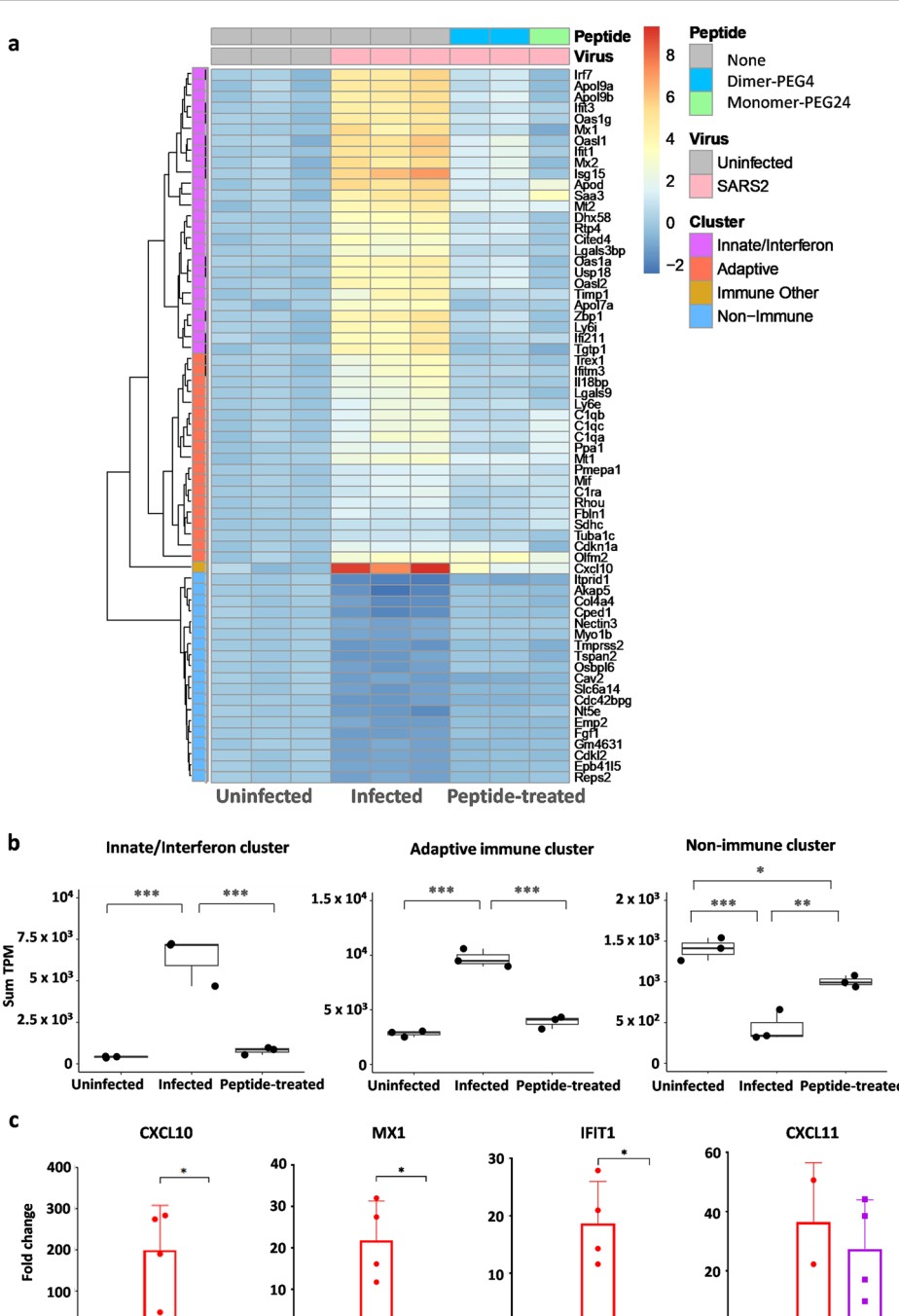

An additional experiment was performed where SARS-CoV-2 Alpha-UK-infected mice were euthanized at earlier time points, 2 and 5 dpi to analyze the viral load in a time-dependent manner (Fig. S5). Although a decrease in viral load was observed in the lungs of peptide-treated mice 2 dpi, similar to what was observed with SARS-CoV-2 D614 infected mice (Fig. 2), the decrease of the viral load was significant principally in the brain homogenates of peptide-treated mice at day 5 pi, suggesting that brain protection is critical for the effect of peptides on the survival of K18-hACE2 mice.

Taken together, these results demonstrate the protective effect of fusion inhibitory peptides against another SARS-CoV-2 variant and suggest that multiple administration of high-dose peptides does not have any evident adverse effect. A repeated peptide administration protects mice against

lethal SARS-CoV-2 infection even in the presence of peptide-specific antibodies.

### Effect of post-treatment with fusion inhibitory peptides on SARS-CoV-2 infection in K18-hACE2 mice

We next analyzed whether post-exposure treatment with fusion inhibitory peptides controls SARS-CoV-2 infection. Twelve K18-hACE2 mice received dimer-PEG4 lipopeptide by i.n. administration of 20 mg/kg, 8 h after infection with $10^4$ PFU SARS-CoV-2 Alpha-UK. Mice received four additional peptide administrations in 24 h intervals and were monitored for 10 days (Fig. 5a). While mice in the control group continued to lose weight and all succumbed to the infection by 7 dpi, post-infection treatment with fusion inhibitory peptide-protected mice from lethal infection (Fig. 5b, c).

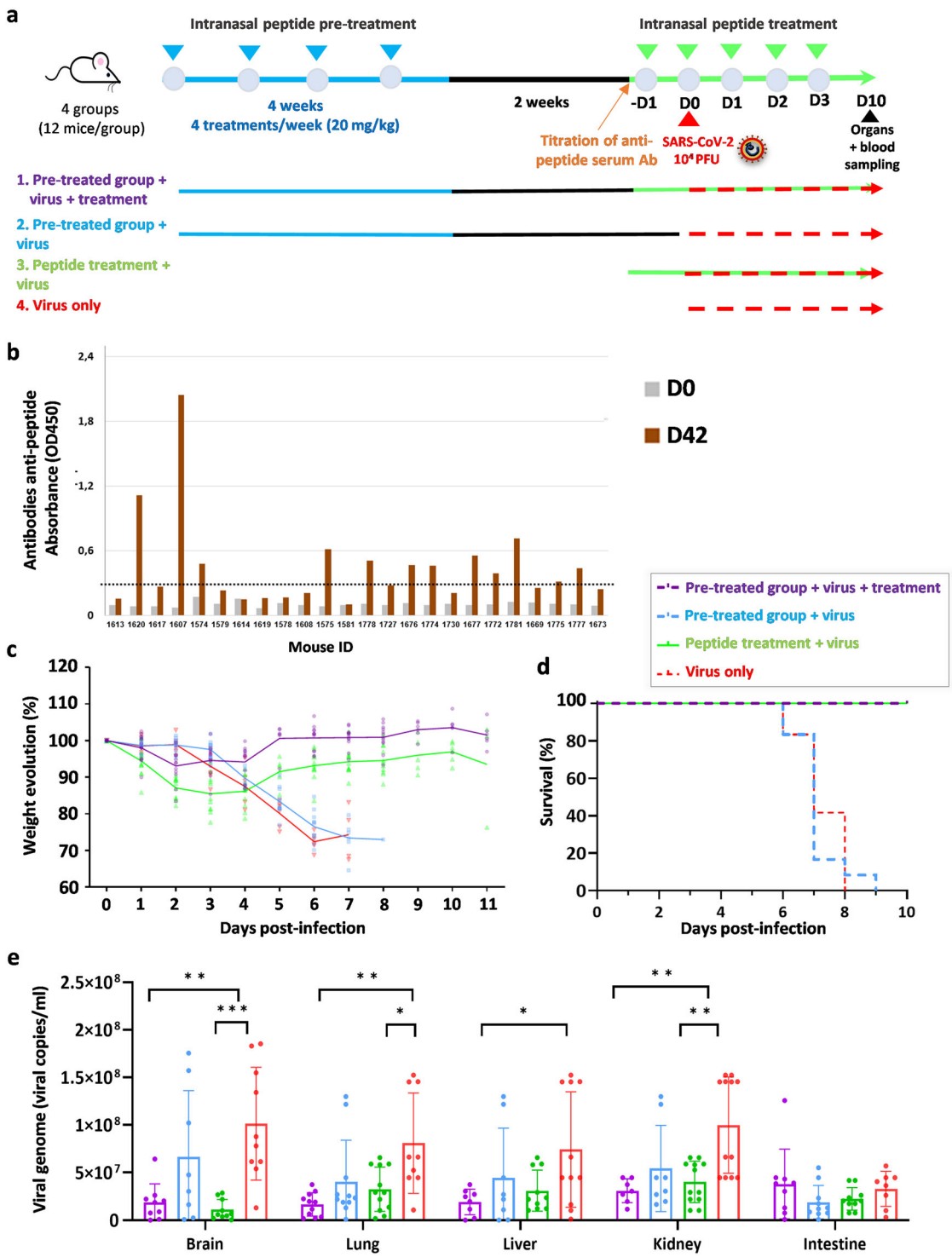

**Fig. 4 | Multiple administrations of high-dose lipopeptides had no adverse effects and did not interfere with peptide-induced antiviral protection during subsequent SARS-CoV-2 challenges. a** Experimental design: 24 K18-hACE2 mice received 20 mg/kg of dimer-PEG4 peptide i.n. 4 times per week, for 4 weeks. After a 2-week break, mice were tested for the presence of peptide-specific antibodies in the serum and separated into 2 groups: one that received the additional peptides (group 1) and the other that remained without treatment (group 2). In addition, 2 groups of naive K18-ACE2 mice (12 mice in each group) were included in the experiment: one that received peptide treatment (group 3), and the other which remained without treatment (group 4). All mice were infected intranasally with SARS-CoV2 alpha-UK ($10^4$ PFU/mouse, in 40 µL) on day 0, and the additional peptide treatment was given intranasally for 5 days (20 mg/kg). Mice were monitored for 10 days and euthanized between days 6 and 9 if they presented clinical signs of infection, and all surviving

mice were sacrificed at day 10, and RNA was isolated from indicated organs. **b** ELISA analysis for the presence of peptide-specific antibodies in the serum of mice from groups 1 and 2, showing the presence of antibodies in 50% of treated mice. **c, d** Weight and survival of mice, after challenge with a lethal dose of SARS-CoV-2 Alpha-UK ($10^4$ PFU). Mantel-Cox Log Rank test between groups shows a statistically significant difference between groups surviving 100% infection (groups 1 and 3) and groups succumbing to infection (groups 2 and 4) (****$p < 0.0001$). **e** Organ-specific viral load between 6 and 10 dpi, tested in different organs by RT-qPCR for the presence of SARS-CoV-2 N, and results are shown as average ± SD. The experiment was repeated in Fig. S5 with mice euthanized at the same dpi (one-way ANOVA with individual comparisons between groups analyzed via Tukey's test (*$p = 0.017-0.045$, **$p = 0.001-0.002$, ***$p = 0.0006$).

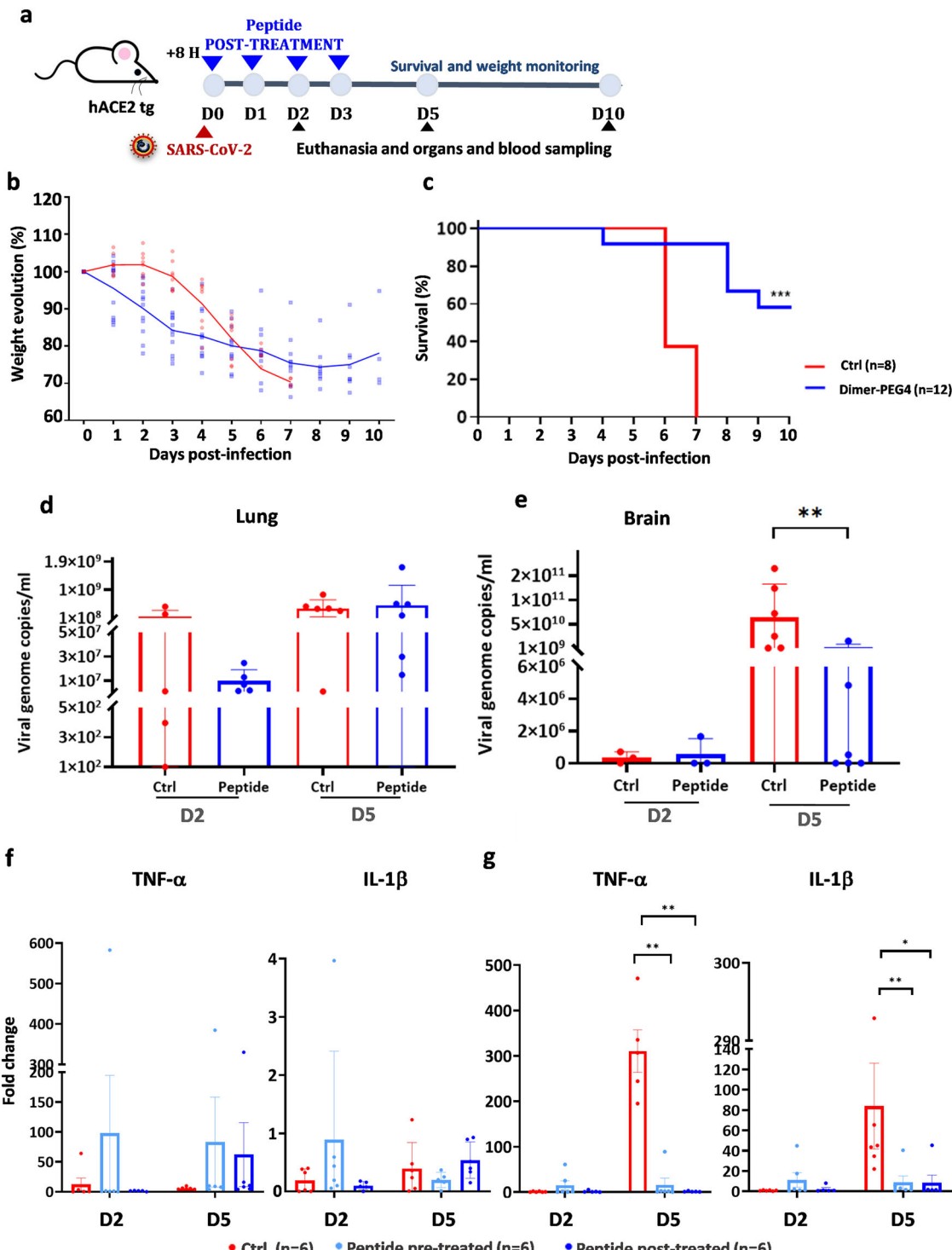

**Fig. 5 | Protective effect of fusion inhibitory lipopeptides given after SARS-CoV-2 infection. a** Experimental design: K18-hACE2 mice were infected i.n. with SARS-CoV-2 alpha-UK ($10^4$ PFU) and received dimer-PEG4 lipopeptide i.n. 8 h later (20 mg/kg), four times in a 24 h interval, and were followed for 10 days. **b, c** Follow up of weight and survival of mice treated with either vehicle (control, ctrl) or dimer-PEG4. Statistical significance of the effect of peptides on survival was measured using the Mantel–Cox test (***$p < 0.0001$). **d** Viral load in lungs in mice euthanized at day 2 pi ($n = 5$) and 5 dpi ($n = 6$). **e** Viral load in the brain of mice euthanized 2 dpi ($n = 3$) and 5 dpi ($n = 6$). Results are presented as mean ± SD. **f, g** RT-qPCR quantification of TNF-α and IL-1ß in lungs (**f**) and brain (**g**) of mice euthanized 2 and 5 dpi. Mice were either pretreated with dimer-PEG4 as described in Supplemental Fig. 5 or post-treated as in (**a**). Results are presented as fold-change (mean ± SD) in comparison to the cytokine expression in noninfected mice. Statistical significance was determined using one-way ANOVA with individual comparisons between groups analyzed via Tukey's test (*$p < 0.05$; **$p < 0.01$).

Similar to results obtained with peptide pre-pretreated mice, RT-qPCR analysis of organs demonstrated reduced viral load in lungs collected at 2 dpi (Fig. 5d), while in brains the viral load was significantly reduced at 5 dpi (Fig. 5e). This reduction was not observed in the lungs at 5 dpi, indicating that the virus titer in lungs at later stages of infection is not the determinant of the infection outcome. These data suggest that post-infection treatment with fusion inhibitory peptides provides significant protection of K18-ACE-2 mice from SARS-CoV-2-induced infection and lethality.

https://doi.org/10.1038/s42003-025-07491-4   **Article**

The notable decrease in viral load in the brains of peptide-treated mice prompted us to search for a possible implicated mechanism. As viral neuroinvasion and encephalitis are considered to be the major cause of death in K18-hACE2 mice[50] and specific cytokines, including IL-1β and TNF-α modulate the blood–brain barrier permeability[51], we analyzed the expression of those cytokines in the organs from control and peptide-treated mice. Although we did not observe differences between untreated and treated mice in cytokine expression in lungs (Fig. 5f), the expression of IL-1β and TNF-α was significantly reduced in brains of peptide pre-treated and post-treated mice at day 5 pi (Fig. 5g). These results suggest that peptide treatment can limit the production of cytokines that are implicated in the permeabilization of blood–brain barrier permeability, decreasing thus virus spread into the brain. This mechanism may contribute to the high survival rate of peptide-treated K18-hACE2 mice following SARS-CoV-2 infection.

### Treatment with fusion inhibitory peptides permits the establishment of long-lasting antiviral immunity against SARS-CoV-2 reinfection

We next analyzed whether SARS-CoV-2 infection in peptide-treated K18-hACE2 mice permits the simultaneous development of adaptive antiviral immunity to protect mice from reinfection. Peptide-treated mice that survived SARS-CoV-2 infection in previous experiments (Fig. 2c) were rechallenged with a lethal dose of SARS-CoV-2 D614 ($10^4$PFU), 21 days after the initial challenge for the dimer-PEG4-treated group (Fig. 6a) and 36 dpi for the monomer-PEG24-treated group (Fig. 6b). All reinfected mice showed a relatively stable body mass and did not develop signs of clinical disease, and all survived the rechallenge with SARS-CoV-2 (Fig. 6a, b).

We next analyzed the development of humoral immunity, by monitoring the production of neutralizing antibodies against SARS-CoV-2 at different time points after infection. All surviving mice in the dimeric peptide-treated group and all except two mice in the monomer-PEG24-treated group produced neutralizing antibodies against SARS-CoV-2 by 14 dpi (Fig. 6c, d). The second challenge with SARS-CoV-2 boosted the production of antibodies, which remained present at a relatively high titer for 6 months post-infection in all dimer-PEG4-treated and 4 months in all monomer-PEG24-treated mice. Although we do not exclude the development of specific T-cell mediate immunity, these results suggested the generation of strong long-lasting humoral immunity in peptide-protected mice.

We next analyzed the production of neutralizing antibodies in mice that had been pretreated with lipopeptides and infected with SARS-CoV-2 Delta variant (Fig. 2e). All peptide-protected mice that survived the infection developed serum-neutralizing antibodies (Fig. 6e). As highly transmissible Omicron variants of SARS-CoV-2 became globally dominant in 2023[52], we evaluated the neutralizing activity of the serum of peptide-protected mice against this variant (Fig. 6g). As Omicron variants are less efficient in the induction of syncytia formation[53], important for sero-neutralization assays, we used vesicular stomatitis virus (VSV) deficient for envelope glycoprotein G and pseudotyped with either Wuhan S protein or the XBB S protein, and used them for the sero-neutralization assay. Sera from peptide-protected mice infected with Delta variant efficiently neutralized VSV pseudotyped with D614G spike (Fig. 6f) or with Omicron XBB spike (Fig. 6g). These results suggest that treatment with fusion inhibitory lipopeptides not only protects mice from a lethal infection with SARS-CoV-2 and from subsequent re-infections, but also allows surviving mice to develop a long-lasting adaptive antiviral immunity, with antibodies that neutralize recently circulating variants.

### Discussion

SARS-CoV-2, like other RNA viruses, mutates rapidly to escape the host immune system[54]. While several vaccines have been developed to combat the virus, only a few antiviral agents have been approved by the US Food and Drug Administration (FDA). New variants have emerged with mutations that either increase receptor-binding affinity or allow the virus to partially overcome the humoral response induced by vaccines[55–58]. Therefore, the development of antiviral approaches that target different steps of the viral cycle, as for other emerging RNA viruses such as HIV, is critical. A synergistic combination of these approaches with vaccination is likely important for controlling future outbreaks.

Peptides derived from the HRC domain of the spike protein are a promising antiviral strategy based on blocking the process of membrane fusion, the critical step for virus entry into host cells. In contrast to the S1 domain of the spike, which contains the receptor-binding region and is prone to frequent mutations, the HRC is located within the S2 domain, which is much more conserved in newly emerging SARS-CoV-2 variants[33]. HRC-derived lipopeptides have demonstrated their effectiveness as antivirals for other virus infections in animals, including human parainfluenza virus type 3, measles virus, and Nipah virus[31,39,40,59], and enfuvirtide (T20 or Fuzeon) is used for the treatment of HIV-1 in humans[60,61]. Fusion inhibitory lipopeptides derived from the SARS-CoV-2 S successfully inhibited S-induced membrane fusion mediated by SARS-CoV and MERS-CoV as well as SARS-CoV-2 and its variants of concern and blocked infection in vitro[18,21,22,62]. These peptides completely prevented SARS-CoV-2 direct-contact transmission in ferrets[20]. The fusion pathways of SARS-CoV-2 variants have evolved, with Omicron variants favoring endo-lysosomal membrane fusion instead of TMPRSS2-driven fusion seen in earlier strains[63]. Despite ongoing debates about the Omicron fusion pathway[64], our previous study confirms that both tested peptides retain antiviral activity, including the Omicron variant and its derivatives BA.1 and BA.2[22].

We expanded here upon previous studies by investigating lipopeptide protection from SARS-CoV-2 lethal infection in a murine model, K18-hACE2 transgenic mice[37], and studying the development of antiviral immune response in peptide-protected mice. As in previous reports[20–22], HRC-derived lipopeptides blocked S-induced membrane fusion and SARS-CoV-2 infection, more effectively than HRN-peptides[65]. In line with the previous observations showing inhibition of SARS-CoV-2 spread in human airway epithelial (HAE) cultures[21], we have shown here that lipopeptides inhibit the replication of SARS-CoV-2 D614 and its Delta variant in lung organotypic cultures from K18-hACE2 mice, and protect mice from fatal SARS-CoV-2 infection. These results support HRC-derived lipopeptides as a promising strategy that is effective not only in preventing virus spread across the population but also in protecting against SARS-CoV-2 infection. This aspect is particularly important for defending vulnerable individuals from severe COVID-19 disease. Post-infection treatment with intranasal lipopeptides protects from infection, although not as well as pretreatment, supporting the feasibility of using lipopeptides for a short period after infection.

SARS-CoV-2 infection of untreated K18-hACE2 mice elicited high pro-inflammatory cytokine, chemokine, and interferon-stimulated gene (ISG) response in lungs, which was reduced to the level of uninfected lungs in the mice pretreated with fusion-inhibitory lipopeptides. SARS-CoV-2-induced the expression of elevated levels of CXCL10, a chemokine known to be associated with a severe course and disease progression in humans and suggested to be a predictive biomarker of outcome in COVID-19[66]. CXCL10 expression was highly diminished in the lungs of lipopeptide-pretreated animals. Cytokine and chemokine production may increase blood–brain barrier permeability, which likely facilitates entry of coronaviruses into the brain of K18-hACE2 mice, contributing to lethality[67]. Indeed, expression of IL-1β and TNF-α, known to modulate blood–brain barrier permeability[51] was significantly decreased in the brains of both lipopeptide pre-treated and post-treated mice, suggesting that peptide treatment reduced SARS-CoV-2 entry into the brain at the later stages of infection and decreased consequent lethality. The absence of an antiviral effect in the brain at 2 dpi may be attributed to early viral spread to the brain through nasal entry, likely reaching the olfactory bulb, as shown in hamster models[68]. These results may explain the high survival rate observed in peptide-treated mice despite the fact that viral load was not eliminated in the lungs of infected animals. Taken together, these data suggest that pretreatment with fusion-inhibitory peptides reduces the initial viral inoculum, allowing the immune system to respond with a more balanced proinflammatory response, avoiding self-damage and promoting better control of infection.

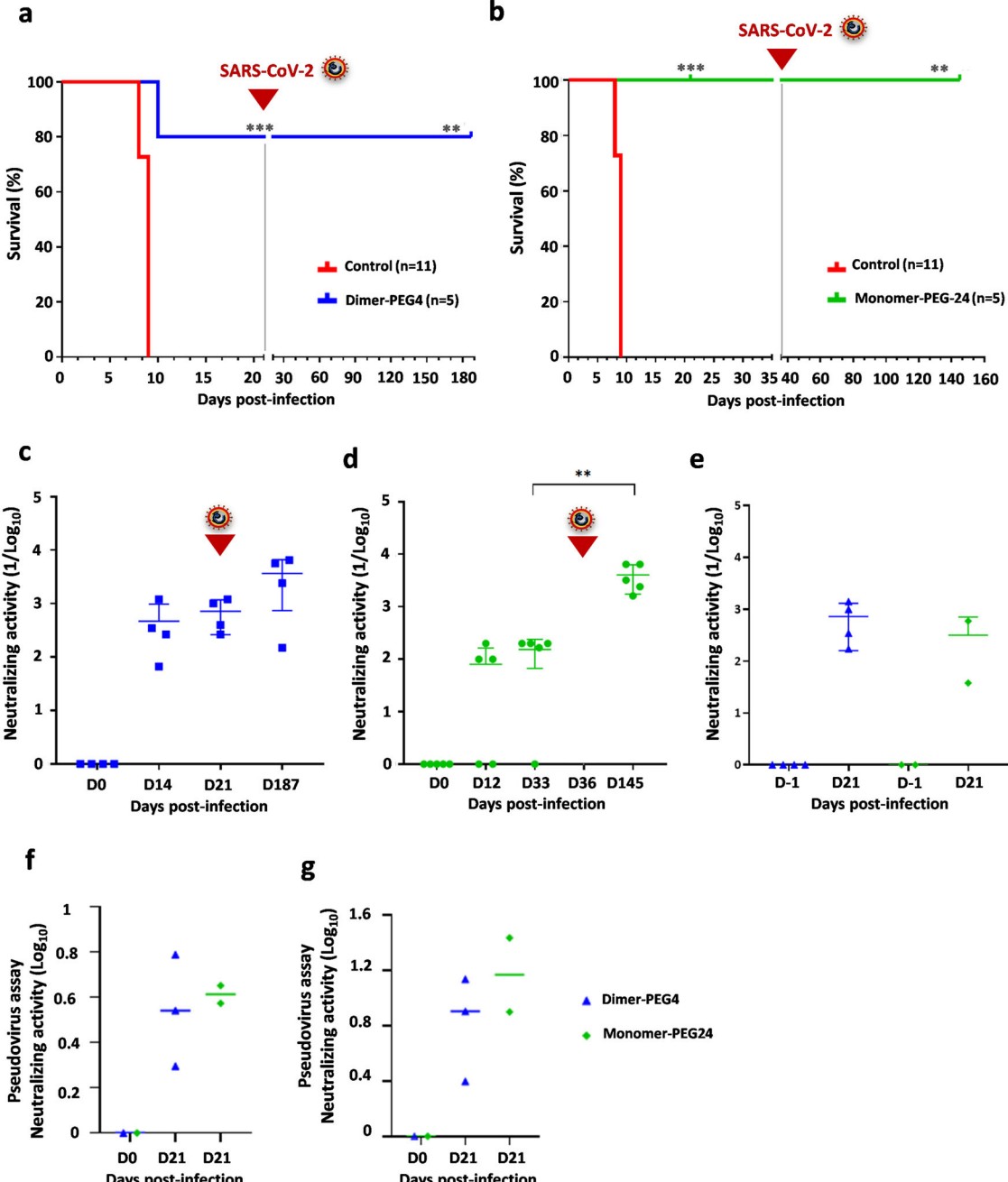

**Fig. 6 | Treatment with fusion-inhibitory lipopeptides allows protected mice to develop a long-term immune response against SARS-CoV-2 variants and resistance to subsequent viral reinfection.** Peptide-treated K18-hACE2 mice from the experiment in Fig. 2 were reinfected **a** 21 days or **b** 36 days after the first infection with the lethal dose of SARS-CoV-2 D614 ($10^4$ PFU) and followed for 6 months. Presence of serum neutralizing antibodies in either dimer-PEG4-treated ($n = 4$) (**c**) or monomer-PEG24-treated mice ($n = 5$) (**d**) at different time points after initial infection with SARS-CoV-2 D614 and reinfection (**p < 0.01, ***p < 0.001,

Mantel–Cox test). **e** Presence of the serum-neutralizing antibodies in either dimer-PEG4-treated (blue, $n = 4$) or monomer-PEG24-treated mice (green, $n = 2$) at 21 days after infection with SARS-CoV-2 Delta. **f, g** Neutralizing capacity of mouse sera was analyzed using the pseudo-VSV system. **f** Ability of sera from mice infected with SARS-CoV-2 Delta (presented in **e**) to neutralize pseudotyped VSV-D614 and **g** to neutralize pseudotyped VSV-Omicron XBB (horizontal bars present mean values).

Administration of high-dose lipopeptides had no adverse effects in mice and did not induce an anti-peptide response that interfered with protection during subsequent SARS-CoV-2 challenge. We showed that intranasal administration of a high dose of the peptides (20 mg/kg) 4 times a week for 4 weeks induced the development of peptide-specific serum antibodies in 50% of treated mice. Although mucosal antibodies present in bronchoalveolar fluid were not analyzed, the generation of specific IgG in the serum did not interfere with the subsequent peptide-induced protection

of mice against the SARS-CoV-2 challenge (Fig. 5c). Repeated administration of antivirals may lead to the generation of escape variants, which is an important concern for antiviral efficacy[69]. However, deep sequencing analysis performed in a previous study revealed no fixed mutations in viruses emerging from SARS-CoV-2 HRC lipopeptide-treated airway tissues[21], suggesting that in these experimental conditions resistance mutations do not emerge under the selective pressure of lipopeptides, similarly to what we have observed in lipopeptide-treated measles virus infections[43].

Finally, we found that intranasal pretreatment with fusion inhibitory peptides followed by SARS-CoV-2 infection not only protected mice from this viral infection but also permitted them to develop an adaptive humoral immune response that protected them against a second challenge by the virus. Serum neutralizing antibodies were detected up to 6 months from the original SARS-CoV-2 infection in the dimer-PEG4-treated group, and up to 4 months in the monomer-PEG24-treated group. As the average lifespan of laboratory mice ranges between 18 and 24 months, this period of immunization is considered long-term, lasting antiviral immunity. Serum from lipopeptide-protected mice neutralized not only the original SARS-CoV-2 variant used for infection but also more recently emerged variants, such as Omicron XBB, suggesting induction of cross-reacting humoral immunity.

Results presented here support the potential of clinical development of SARS-CoV-2 lipopeptide antivirals for humans. Indeed, two SARS-CoV-2 fusion inhibitory lipopeptides are currently evaluated in clinical phase II/III trials via nebulization or aerosol administration[70,71]. We have previously shown that fusion-inhibitory peptides can be administered by aerosolization to inhibit infection with the measles virus, known to be one of the most contagious viral diseases[43]. Nebulization of our SARS-CoV-2 lipopeptides should be further evaluated in clinical assays and may provide new options for the prevention of infection. Our findings, along with previous studies provide a proof of concept for the use of fusion-inhibitory peptides as an antiviral approach against COVID-19, as well as the basis of a rapid emergency response that can be implemented upon identification of a potentially pathogenic enveloped virus.

## Methods

### Cell lines, viruses, plasmids, and peptides

Human embryonic kidney (HEK) 293T (CRL-3216™) and Vero E6 (African green monkey kidney, CRL-1586™) cells, both derived from female individuals, were obtained from the American Type Culture Collection (ATCC, USA), and grown in Dulbecco's modified Eagle's medium (DMEM GlutaMax, Invitrogen, Carlsbad, CA, USA) supplemented with 10% fetal bovine serum (FBS) and antibiotics (100 IU/mL of penicillin and 100 μg/mL of streptomycin) in 5% $CO_2$ incubators at 37 °C and were tested negative for *Mycoplasma* spp. (MycoAlert, Lonza LT07-318).

Recombinant SARS-CoV-2 virus expressing the mNeonGreen reporter gene inserted into the ORF7 of the viral genome, SARS-CoV-2-mNG, was kindly provided by Dr. Vineet D. Menacherry and Dr. Pei-Yong Shi (University of Texas Medical Branch, Galveston, TX, USA) and was used only for organotypic culture infections and imaging. BetaCoV/France/IDF0571/2020 virus (GISAID Accession ID = EPI_ISL_411218) was kindly provided by Virpath lab (CIRI, Lyon, France), and Delta variant GISAID: EPI_ISL_3431813 was provided by Dr. A.G. Marcelin. These viruses, as well as South African beta variant B.1.351 (SA; SARS-CoV-2 strain hCoV-19/USA/MD-HP01542/2021) (BEI Resources NR-55282), were used in the in vitro and ex vivo studies. The SARS-CoV-2 strain BetaCoV/France/IDF0372/2020 used in the in vivo studies was kindly provided by Pr. Y. Yazdanpanah and the French National Reference Centre for Respiratory Viruses hosted by the Institute Pasteur (Paris, France). In addition, the UK alpha variant B.1.1.7 (UK; SARS-CoV-2 strain hCoV-19/England/204820464/2020) (BEI Resources NR-54000) and Omicron variant hCoV-19/USA/WA-UW-TC-21113017867/2021 (GISAID: EPI_ISL_9142381) were used for in vivo experiments. All viruses were produced by infecting Vero E6 cells with a MOI of 0.02 in DMEM. After 1 h of virus adsorption at 37 °C, the medium was replaced with DMEM at 2% FBS, and the cultures were incubated at 37 °C in 5% $CO_2$ for three days. Virus supernatant fluid was then collected and cleared by centrifugation at $500 \times g$ for 5 min, aliquoted, and then titrated as plaque-forming units using the limiting dilution method and supernatants were stored at −80 °C.

The genes for the SARS-CoV-2 Spike (Wuhan strain) and the human Angiotensin-Converting Enzyme 2 (hACE2) proteins were codon optimized, synthesized, and subcloned into the mammalian expression vector pCAGGS and then used for the β-Gal complementation-based fusion assay.

Both lipid-tagged SARS-CoV-2 inhibitory peptides, monomer-PEG24 (SARS$_{HRC}$–PEG$_{24}$-chol) and dimer-PEG4 ([SARS$_{HRC}$–PEG$_4$]$_2$-chol) are HRC-peptide derivatives corresponding to residues 1168−1203 of SARS-CoV-2 S (Wuhan strain) with a C-terminal -GSGSGC spacer sequence, and were prepared by solid-phase peptide synthesis and conjugated to PEG-cholesterol linkers as described previously[20–22]. As a control, lipopeptide derived from the HRC domain of the F protein of measles virus (HRC4)[43] was used.

### β-Gal complementation-based fusion assay

The β-galactosidase (β-Gal) complementation-based fusion assay was performed as described previously[20,21]. Briefly, the hACE2 receptor-bearing 293T cells expressing the omega reporter subunit of β-Gal were incubated at 37 °C overnight with cells co-expressing SARS-CoV-2 S and the alpha subunit of β-Gal in the presence of different concentrations of the peptides. Alpha-omega complementation indicates cell fusion and is revealed by the addition of substrate (®The Tropix Galacto-Star™ chemiluminescent reporter assay system, Applied Biosystem) after lysing the cells. A Tecan M1000PRO microplate reader was used for luminescence quantification. The percentage of inhibition was calculated as the ratio of relative luminescence units in the presence of a specific concentration of inhibitor and the relative luminescence units in the absence of inhibitor and corrected for background luminescence as follows: percent inhibition $=100\times[1-($ luminescence at X−background)/(luminescence in the absence of inhibitor−background)].

### Viral entry assay

Vero E6 monolayer cells were incubated with SARS-CoV-2 at a MOI of $10^{-4}$ in the presence of different concentrations of peptides. Peptides were diluted in DMEM 4% FBS, added to the cells after removing the medium, and incubated for 30 min. Virus resuspended in DMEM was added to the peptide solution and incubated with cells for 90 min, and the medium was replaced with fresh DMEM 4% FBS mixed with 2% carboxymethyl cellulose medium viscosity as an overlay medium. The plates were incubated for 4 days at 37 °C in 5% $CO_2$ and revealed using a crystal violet solution. The titer for each condition was determined by end-point dilution and counting the number of plaque-forming units.

### Preparation of organotypic cultures, SARS-CoV-2 infection, and peptide treatment

Organotypic slices were prepared from K18-hACE2 transgenic mouse lungs and brains as described previously[44]. Briefly, lungs, cerebellum, and olfactory bulbs were isolated from 7- to 10-day-old K18-hACE2 transgenic mice (euthanized by decapitation) and cut with a McIlwain tissue chopper (WPI-Europe) into 500 μm-thick lung slices. For each condition, five slices were prepared from five different mice without sex discrimination. Organotypic slices were then dissociated in cold Hibernate®-A/5 g/L ᴅ-Glucose buffer and then deposited on Millipore hydrophilic polytetrafluoroethylene membranes (Millipore, PICM0RG50). The slices were subsequently cultured in GlutaMAX minimal essential medium supplemented with 25% horse serum, 5 g/L of glucose, and 1% HEPES (Thermo Fisher Scientific), in addition to 1 mg/mL of human recombinant insulin (Sigma-Aldrich). The cultures were maintained at 37 °C in a 5% $CO_2$ atmosphere, and medium change was performed daily after the slicing procedure. Organotypic cultures were infected on the day of preparation with either SARS-CoV-2 D614 or SARS-CoV-2-mNG or SARS-CoV-2 Delta (at 500 PFU). For each virus, viral replication was assessed in the lungs either by RT-qPCR or fluorescence microscopy, respectively. For treatment, slices were treated with the peptides to reach the desired concentration starting 90 min before the infection. The feeding medium was replaced daily with fresh medium containing a concentration of 1 μM of peptides until 4 dpi. In addition, at each time point (90 min before infection, and then 24, 48, and 72 h.p.i.) a 2 μL drop containing the peptide at a final concentration of 10 μM was delicately added at the top of each slice. At 96 h.p.i. the organotypic cultures

infected with SARS-CoV-2 were collected and homogenized with beads in 350 µL of RA1 Lysis buffer (Macherey Nagel) in a Tissue Lyser II (Qiagen) for RNA extraction. At 48 and 72 h.p.i., slices obtained from organotypic cultures were imaged using fluorescence microscopy. Images were scanned and analyzed using QuPath, an Open source software for digital pathology image analysis[72].

## MTT viability test

MTT assay was performed to evaluate the metabolic activity or organotypic cultures following peptide treatment, to assess the possible peptide toxicity. Organotypic cultures were treated for 3 days with either medium, 1 µM lipopeptides, or 0.5% SDS. Metabolic activity was assessed by adding to each slice 70 µL of 3-(4,5-dimethylthiazol-2-yl)-2,5-diphenyltetrazolium bromide (MTT, 5 mg/mL) following with incubation 4 h at 37 °C. Slices were transferred in a plate (one slice per well) with 200 µL of DMSO and heated for 10 min at 37 °C to solubilize formazan. 75 µL of the reaction was transferred, and the absorbance was measured at OD 540 nm (Tristar 5, Berthold).

## Study approval

Studies involving mice were performed in accredited animal biosafety level 3 laboratories: Centar of Immunophenomic, CIPHE, Marseille, France, and at Biovivo, Institut Claude Bourgelat at Marcy l'Etoile, France, and at Columbia University Irving Medical Center, New York, USA. All protocols including handling and manipulating animals received ethical approval from the Regional ethical committee and the French Ministry of High Education and Research (French Animal Regulation Committee Number APAFIS#26484-2020062213431976-v6 for CIPHE and v6APAFIS#27797-2020100516408472 for Biovivo) and by the Institutional Animal Care and Use Committee at Columbia University School of Medicine (animal protocol number AC-AABG9559). We have complied with all relevant ethical regulations for animal use. Virus inoculations were performed under anesthesia, and all efforts were made to minimize animal suffering. Mice were monitored daily for morbidity (body weight) and mortality (survival). During the monitoring period, mice were evaluated for clinical symptoms, and those reaching a clinical score meeting the experimental end-point criteria were humanely euthanized. Animals were scored daily using a quantitative assessment chart for pain and distress, which included parameters such as body weight (0: normal; 1: <10% weight loss; 2: 10−15% weight loss, eating; 3: >20% weight loss, not eating), appearance (0: normal; 1: lack of grooming; 2: coat rough, possible nasal and or ocular discharge; 3: coat very rough, abnormal posture, pupils enlarged), clinical signs (0: normal, 1: small change of potential significance; 2: temperature rise 1–2 °C, mildly labored breathing observing thorax and flanks; 3: temperature change >2 °C, marked labored breathing observing thorax and flanks), unprovoked behavior (0: normal; 1: minor changes; 2: abnormal behavior, less mobile, less alert, inactive when activity expected; 3: unsolicited vocalization, extreme self-mutilation) and response to external stimuli, palpation or injection sites (0: normal; 1: minor exaggerated response; 2: moderate abnormal response; 3: violent response). A score of 3 or less was considered normal, 4–7 indicated mild to moderate discomfort, 8−11 suggested significant suffering requiring intervention, and 12−15 represented severe pain requiring immediate veterinary consultation or euthanasia. Additionally, any single score of 3 in an independent category automatically placed the animal in the 8−15 range. Animals were euthanized if their total score exceeded 8, ensuring humane treatment and adherence to ethical guidelines.

## Mice, infection, treatment, and sampling

Heterozygous K18-hACE C57BL/6J mice (strain: B6.Cg-Tg(K18-ACE2)2Prlmn/J, Jax strain ID: 034860)[67] in the C57BL/6J background, were obtained from Jackson Laboratory and breeding, genotyping, and handling of mice was performed at the institutional animal facilities. Animals were housed in groups and fed standard chow diets. Mice of different ages and both sexes were used in the following way: in Fig. 2b, c, the control group

contained 50% males and 50% females, monomer-PEG24-treated group contained 100% males, and dimer-PEG4-treated group 100% females. In Figs. 2d, e, 4, and 5 were mice included 50% of each sex in all the conditions. Mice were administered intranasally under isoflurane or ketamine/xylazine (100 and 10 mg/kg, respectively) anesthesia with indicated doses of SARS-CoV-2 in the volume of 30–40 µL. In some experiments, mice received the second virus challenge 3–4 weeks after the first infection. Administration of peptides was done intranasally under anesthesia with a mixture of ketamine/xylazine (100 and 10 mg/kg, respectively), using 4 or 20 mg/kg of lipopeptides, depending on the experiment, dissolved in 40 µL of water and 2% DMSO (20 µL each nostril), while control animals received only the vehicle (2% DMSO). In some experiments, oral swabs were taken 2 days after infection, using the cotton sticks soaked in PBS and, after scraping, transferred into tubes containing 400 µL of AVL Qiagen buffer and then kept at −80 °C before RNA extraction. Blood was taken at different time points by retroorbital bleeding during the experiment or by heart puncture at the end of the experiment. Blood was left for 30 min at room temperature, and sera were obtained after centrifugation at $1600 \times g$ for 10 min, aliquoted, and kept frozen before the utilization. Mice were euthanized by cervical dislocation and organs: lungs, brain, liver, kidneys, spleen, and intestines were collected at the end of the experiment.

## ELISA for semi-quantitative peptide measurement

Lungs were harvested, weighed, mixed in PBS (1:1, w/vol), and homogenized using a BeadBugTM microtube homogenizer. Samples were subsequently treated with acetonitrile/1% TFA (1:4, vol/vol) overnight on a rotor at 4 °C and centrifuged for 10 min at 8000 rpm. 96-well plates Maxisorp (Nunc) were coated overnight with purified rabbit anti-HRC-SARS antibodies in carbonate/bicarbonate buffer (pH = 7.4, 20 g/mL). Plates were washed twice in PBS and blocked in 3% BSA/1X PBS for 30 min. The blocking buffer was replaced by two dilutions of each sample in 3% BSA/1X PBS in duplicate and incubated for 1.5 h at room temperature (RT). Wells were washed three times in 1X PBS and developed using purified rabbit anti-HRC-SARS antibodies conjugated to biotin for 1 h at RT. Wells were washed three times in 1X PBS and developed using streptavidin conjugated to peroxidase in 3% BSA/1X PBS for 30 min at RT followed by five washes and incubation with Ultra-TMB substrate solution (Sigma-Aldrich). Reactions were stopped with sulfuric acid (12%) and absorbance was read at 450 nm.

## ELISA for the determination of anti-peptide antibodies

Mice were inoculated i.n. with 20 mg/kg in total volume of 40 µL, 4 times a week for 4 consecutive weeks and serum was collected. 96-well plates Maxisorp (Nunc) were coated overnight with monomer PEG4 peptide in carbonate/bicarbonate buffer (pH = 7.4, 100 ng/100 µL/well). Plates were washed three times in PBS and blocked in 0.5%/1X PBS for 1 h. The blocking buffer was replaced by 4 dilutions of each sample in 0.5%/1X PBS and incubated for 1 h at RT. Wells were washed three times in 1X, and the ELISA was developed using HRP-conjugated antibody (Goat anti-Ms IgG, #115-035-003, Jackson ImmunoResearch, 1:5000) in 0.5% milk for 1 h at RT, followed by three washings of 1X PBST and incubation with Ultra TMB substrate solution (Sigma-Aldrich). The reaction was stopped with sulfuric acid (12%), and absorbance was read at 450 nm.

## RNA extraction and real-time RT-PCR

Murine organs were collected, weighed, and conserved in RNA later buffer (QIAGEN). RNA was extracted using the RNeasy Mini-Kit (QIAGEN) and reverse transcribed using the High-Capacity cDNA Reverse Transcription Kit (Thermo Fisher Scientific). Amplification was carried out using One-Green Fast qPCR Premix (OZYME) according to the manufacturer's recommendations. The number of copies of the SARS-CoV-2 RNA-dependent RNA polymerase (RdRp) gene or nucleoprotein (N) gene in samples was determined using the primers indicated in Table 1. The amplified region was included in a cDNA standard to allow copy number determination down to 100 copies per reaction. The copies of SARS-CoV-2

**Table 1 | Oligonucleotides used for RT-qPCR (sequence 5'–3')**

| Gene | Forward primer | Reverse primer |
|---|---|---|
| SARS-CoV-2 Nucleoprotein | AAACATTCCCACCAACAG | CACTGCTCATGGATTGTT |
| SARS-CoV-2 RNA-dependent RNA polymerase | CATGTGTGGCGGTTCACTAT | GTTGTGGCATCTCCTGATGA |
| Mouse GAPDH | GCATGGCCTTCCGTGTCC | TGTCATCATACTTGGCAGGTTTCT |
| Mouse CXCL-10 | GGTCTGAGTCCTCGCTCAAG | GTCGCACCTCCACATAGCTT |
| Mouse CXCL-11 | TATGTTCAAACAGGGGCGCT | ACTTTGTCGCAGCCGTTACT |
| Mouse MX1 | TCCATTGGTCTTCTGTCACCCG | AGACCATCCACCTCCACTTCTC |
| Mouse IFIT1 | CATGTTGAAGCAGAAGCACACA | TTGGCTGCATAGCGAATGAC |
| Mouse IFN gamma | ATTCAGAGCTGCAGTGACCC | ACATTCGAGTGCTGTCTGGC |
| Mouse TNF alpha | CTGTAGCCCACGTCGTAGC | TTGAGATCCATGCCGTTG |
| Mouse IL-1 beta | GACCTTCCAGGATGAGGACA | AGGCCACAGGTATTTTGTCG |

were compared and quantified using a standard curve and normalized to total RNA levels. An external control (mock-infected animal, nondetectable in the assay) and positive control (SARS-CoV-2 cDNA containing the targeted region of the RdRp gene at a concentration of $10^4$ copies/µL) were used in the RT-qPCR analysis to validate the assay. To quantify GAPDH (Glyceraldehyde 3-phosphate dehydrogenase) transcription, as well as cytokines and chemokines expression, 100 ng of total RNA was reverse-transcribed using iScript cDNA Synthesis Kit (Bio-Rad) according to the manufacturer's instructions in presence of oligo-dT and random hexamer primers. The cDNAs were then diluted to 1-to−10, and 5 µL of each diluted cDNA was finally used as the template for qPCR using Platinum SYBR Green qPCR SuperMix-UDG kit (Invitrogen) on a StepOnePlus Real-Time PCR System (Applied Biosystems). All primers used in the study had an efficacy close to 100% according to the MIQE checklist. All samples were run in duplicates, and results were analyzed using StepOne version 2.3 (Applied Biosystems). The qPCR results of cellular transcripts were normalized and expressed as a number of copies of the target gene RNA per indicated weight of tissue, based on the standard deviation (SD) for GAPDH mRNA. Fold changes were determined using the $2^{-\Delta\Delta Ct}$ method, as described previously[73].

### Transcriptomic analysis

RNAs were extracted from mouse lungs using NucleoSpin RNA Kit (Macherey-Nagel). Total RNA was quantified by spectrophotometry under a DS-11-FX (DeNovix) and submitted to the JP Sulzberger Columbia Genome Center for library preparation and sequencing. Strand-specific RNA-Seq libraries were prepared using a poly-A enrichment (TruSeq Stranded mRNA kit; Illumina) and were sequenced on an Illumina NovaSeq 6000 with paired-end 2x100 reads as described previously[44]. Paired-end reads were pseudo-aligned to the mouse transcriptome GRCm38 with Kallisto v0.44 then all transcripts collapsed to gene level prior to differential expression (DE) analysis[74]. Raw count data were filtered to remove genes averaging fewer than 10 counts per sample. Expression means were normalized and fitted to a negative binomial distribution, and the Wald test for differential expression was performed using DESeq2 v1.34.0[75] in R v4.1.0. Out of a total pool of 34,945 genes, 15,423 remained after low-count filtering and were tested for DE.

The 65 genes with the lowest adjusted $p$-value in the DE analysis (range: $1.24^{-38}$–$4.4^{-13}$) between the untreated SARS-CoV-2 infected and uninfected groups of mice were selected for display, and log$_2$-fold change in each treatment group relative to the uninfected control was plotted with the R package pheatmap. To control for the effect of transcript length, transcripts per million (TPM) values were used to determine expression sums per gene category. Raw sequencing reads have been deposited to the NCBI Sequence Read Archive (GEO under accession GSE223056).

### Immunohistochemistry

Lungs were harvested from the euthanized infected K18-hACE2 mice and uninfected controls and fixed in 4% paraformaldehyde solutions. Tissues were put in a 20% sucrose solution for two days at 4 °C and then immersed in iso-pentane under dry-ice before inclusion in cryo-embedding media (OCT). Specimens were stored at −80 °C and then cut using a Cryostat (Leica CM1950) to 7 µm diameter sections, and placed on positively charged glass slides (Superfrost). Slides were dried for several hours at 47 °C before being stored at −80 °C. For staining, after inhibition of endogenous peroxidase activity, non-specific epitopes were blocked with PBS supplemented with 3% bovine serum albumin (BSA) and 0.15% Triton X-100 for 30 min. Slides were stained by primary rabbit anti-SARS-CoV-2 N antibody (NOVUSBIO, NB100-56576) diluted at 1/2000 and incubated overnight at 4 °C in the blocking buffer, before adding the secondary anti-rabbit Ig/peroxidase (PROMEGA, W401B) diluted at 1/500. Peroxidase substrate (ImmPACT AMEC-Red from VECTOR ref SK-4285) for 9 min to reveal the signal and tissues were counterstained with hematoxylin QS (VECTOR ref H-3404). Slides were mounted with an aqueous medium and imaged using a Zeiss Axiovert 100M microscope.

### Seroneutralization assay

Serum was kept at −80 °C, thawed before analysis, diluted 25× in 50 µL DMEM (0% FBS), and mixed with 200 PFU of SARS-CoV-2 D614 resuspended in 50 µL of DMEM. The mix was then incubated for 30 min at RT in a 96-well plate in quadruplicate for each sample. 100 µL of DMEM supplemented with 4% FBS containing $4 \times 10^4$ Vero E6 cells was added to the mix in each well in order to have a final concentration of serum of 1:100–1:12,800. After 4 days of incubation at 37 °C in 5% CO$_2$, the cytopathic effect was revealed by crystal violet staining, and the neutralization endpoint titers were expressed as the last dilution of serum that completely abolished SARS-CoV-2 infection and cytopathic effect.

### Generation of pseudotyped SARS-CoV-2 VSVΔG and evaluation of sero-neutralization

We generated recombinant vesicular stomatitis virus (VSV) lacking the envelope glycoprotein G gene—rVSVΔG-RFP and rVSVΔG-GFP, pseudotyped with two different SARS-CoV-2 S proteins: D614G using pCAGGS-puro plasmid[21], and Omicron XBB strain in pCAGGS SARS-CoV-2 XBB Spike plasmid (gift from Marceline Côté, Addgene plasmid# 195287; http://n2t.net/addgene:195287; RRID:Addgene_195287). In these recombinant constructs, the sequence coding for VSV attachment protein G was deleted and replaced by either reporter red fluorescent protein (RFP) or green fluorescent protein (GFP) sequences giving rVSVΔG-R/GFP, as previously described[76]. The complementation with SARS-CoV-2 glycoproteins expressed in cells in trans enabled us to produce stocks of pseudotyped viruses identical in their genetic background that differed only in the nature of the surface glycoproteins decorating viral particles. As the infectivity of rVSVΔG pseudotypes is restricted to a single round of replication, they have been used for the study of viral entry for a broad range of highly pathogenic viruses[77].

For pseudotyping, spike protein-encoding plasmids were transfected into HEK293T cells by using JetOptimus Transfection Reagent (polyplus). Cells were infected with rVSVΔG-RFP-VSVG[76] 8 h post-transfection. After 24 h, the supernatant fluid was clarified for 5 min at $2500 \times g$ and then subjected to ultracentrifugation for 90 min at $96,000 \times g$ on a sucrose cushion in a Beckman SW32 rotor. Replicons pseudotyped with SARS-CoV-2 spike proteins were mixed with diluted sera (1/20–1/1260) and incubated for 30 min at 37 °C. After incubation, the mix was inoculated on adherent VeroE6 cells for 8 h. The percentage of infected cells was determined by flow cytometry (BD LSRFortessa5L, BD Biosciences) after detaching the cells with trypsin. The serum neutralization dilution for each analyzed serum was determined using GraphPad Prism8 non-linear regression and analysis mode: [Agonist] vs. response-variable slope (four parameters).

## Statistics and reproducibility

We used one-way ANOVA analysis, Tukey's test, and two-tailed Mann–Whitney test for statistical analyses of results of the inhibition of infection and the quantification of gene copies and Mantel–Cox for analysis of animal survival. We considered p-values of 0.05 or below to be statistically significant. Inhibitory concentration 50% ($IC_{50}$) was calculated using non-linear fit regression and "[Inhibitor] vs. response—variable slope (four parameters)" model. $IC_{50}$ calculations and statistical analyses were performed using GraphPad Prism 8.3 software.

The number of animals used in each presented experiment was selected to allow meaningful statistical analyses and indicated in the figure legends and individual values presented in Supplementary data files. The animals were randomly assigned to the experimental groups by simple randomization handled by personnel not directly involved in the study (animal facility personnel). Experimenters were blinded to group assignments during virus inoculation, and treatment (vehicle vs. peptides). Blinding was also applied during the in vitro experiments and analyses. All in vitro experiments were successfully replicated three times, consistently yielding similar results. In vivo pre-treatment experiments involving SARS-CoV-2 were reproduced using different SARS-CoV-2 variants (Wuhan, alpha, delta), further reinforcing the reproducibility of the data.

In the transcriptome analysis, the 65 genes with the lowest adjusted p-value (range: $1.24^{-38}$–$4.4^{-13}$) between the untreated SARS-CoV-2 infected and uninfected groups of mice were selected for display, and $log_2$-fold change in each treatment group relative to the uninfected control was plotted with the R package heatmap. Results presenting adaptive/innate/interferon metadata bars are based on the non-redundant lists of genes from reactome+GO BioProcess. The sum of normalized counts per sample was determined and results are presented for each of 3 analyzed clusters by boxplots showing individual values and the average of the normalized gene expression for each group. Comparison between uninfected, infected-untreated, and infected-peptide-treated groups was performed using ANOVAs with Tukey's post hoc test.

## Reporting summary

Further information on research design is available in the Nature Portfolio Reporting Summary linked to this article.

## Data availability

Raw sequencing reads have been deposited to the NCBI Sequence Read Archive (GEO under accession GSE223056). Values for all data points presented in graphs are reported in the Supplementary data file. Requests for further information on data availability or for resources and reagents should be directed to co-corresponding authors M.P. and B.H.

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

## Acknowledgements

We acknowledge World Reference Center for Emerging Viruses and Arboviruses (WRCEVA) and UTMB investigator, Dr. Pei Yong Shi for kindly providing recombinant NeonGreen virus based on 2019-nCoV/USA_WA1/2020 isolate and VirPath from the CIRI, Centre International de Recherche en Infectiologie, Lyon, France (B. Lina, A. Pizzorno, O. Terrier, and M. Rosa-Calatrava), for providing us with the BetaCoV/France/IDF0571/2020 virus. We are grateful to all the members of the group Immunobiology of viral infection at CIRI, as well as Centre d'Immunophénomique, Marseille and Biovivo, VetagroSup, Marcy l'Etoile, France, to Lokendrasingh V. Chauhan from Columbia University, USA for the help during the performance of this study and M. Ledevin and T. Larcher from Oniris for the help with IHC approach. This work was supported by ANR-CoronaPepStop (ANR-20-COVI-000) and by funding from the National Institutes of Health (RO1 AI160961 to A.M. and RO1 AI160953 to M.P.).

## Author contributions

B.H., M.P., A.M., A.Z., G.N. and S.Mo. designed and supervised the research studies. S.Mo., V.F., C.P., O.R., S.D., M.M., C.C., E.P., D.D., F.T.B., L.M.L., N.A.P.L., C.M., G.N., A.Z. and S.Ma. conducted experiments and acquired data. S.Mo., V.F., C.P., C.M., B.M., A.L.G., N.V.D., A.G.M., A.M., M.P. and B.H. analyzed data. C.A.A. performed peptide analysis. S.Ma., T.B., and A.G.M. provided biological reagents. S.Mo. and B.H. wrote the manuscript. N.A.P.L., C.M., A.L.G., B.M., A.Z., A.M., M.P. and B.H. edited the manuscript. S.Mo. being more involved in the design and analysis of data and V.F. and C.P. in conducting the experiments.

## Competing interests

M.P. and A.M. anticipate future financial interest in Thylacine Biotherapeutics, a company established to develop lipopeptide antiviral therapeutics. A.M., M.P., B.H. and C.M. are inventors of several provisional patent applications related to the use of antiviral lipopeptides.

## Additional information

¹CIRI, Centre International de Recherche en Infectiologie, Inserm, U1111, Université Claude Bernard Lyon 1, CNRS, UMR5308, Ecole Normale Supérieure de Lyon, Lyon, France. ²Département du Rhône, Lyon, France. ³Division of Pediatric Critical Care Medicine and Hospital Medicine, Department of Pediatrics, Vagelos College of Physicians and Surgeons, Columbia University Irving Medical Center, New York, NY, USA. ⁴Department of Chemistry, Materials and Chemical Engineering "G. Natta and Department of Electronics, Information and Bioengineering, Politecnico of Milan, Milan, Italy. ⁵Institute of Comparative Medicine, Columbia University Irving Medical Center, New York, NY 10032, USA. ⁶Department of Laboratory Medicine and Pathology, University of Washington Medical Center, Seattle, WA, USA. ⁷Institut Claude Bourgelat, VetAgro Sup, Marcy l'Etoile, Lyon, France. ⁸Centre d'Immunophénomique, Aix Marseille Université, Inserm, CNRS, PHENOMIN, Celphedia, Marseille, France. ⁹Center for Infection and Immunity and Department of Epidemiology, Mailman School of Public Health, Columbia University, New York, NY, USA. ¹⁰Robert Frederick Smith School of Chemical and Biomolecular Engineering, Cornell University, Ithaca, NY, USA. ¹¹Sorbonne Université, Virology department, Pitié-Salpêtrière hospital, AP-HP, Pierre Louis Epidemiology and Public Health institute, INSERM 1136 Paris, France. ¹²Center for Host-Pathogen Interaction, Vagelos College of Physicians and Surgeons, Columbia University Irving Medical Center, New York, NY, USA. ¹³Department of Microbiology & Immunology and Department of Physiology & Cellular Biophysics, Vagelos College of Physicians and Surgeons, Columbia University Irving Medical Center, New York, NY, USA. ¹⁴Department of Experimental Medicine, University of Campania Luigi Vanvitelli, Caserta, Italy. ¹⁵These authors contributed equally: Said Mougari, Valérie Favède, Camilla Predella. ✉e-mail: mp3509@cumc.columbia.edu; branka.horvat@inserm.fr

