## [Transparent Peer Review file · Communications Biology]

Intranasally administrated fusion-inhibitory lipopeptides block SARS-CoV-2 infection in mice and enable long-term protective immunity

Corresponding Author: Dr Branka Horvat

Version 0:

Reviewer comments:

Reviewer #1

(Remarks to the Author)

The paper by Mougari et al describe a lipopeptide formulation derived from HRC domain of S inhibit viral infection in vitro and show efficacy in both prophylaxis and post exposure prophylaxis in the K18ACE2 model of SARS-CV2 infection. Interestingly, the peptides do not interfere with convalescent anti-viral antibody response. Overall, the experiments are well performed and the data support the author's conclusions. There are some points that need to be clarified.

1. The 20 uL volume of peptide delivery will result in the material being aspirated into the lung as demonstrated by the authors. Thus the current dataset does not clarify whether an intranasal formulation or an inhaled formulation would be needed to move this program to the clinic. Do the authors have data with smaller volumes, such as 10 uL which have been shown to reduce lung delivery? Some of these points about formulation and route of delivery should also be added to the Discussion.
2. The authors perform repeated dosing to assess toxicity but more relevant would be to assess olfaction in mice (<https://pubmed.ncbi.nlm.nih.gov/31641151/>) to make sure the peptides are not toxic to olfactory neurons.
3. In Figure 2, why is there no effect of the peptide on CNS viral load at day 2?
4. For the organotypic cultures, were the lungs inflated with LMP agarose similar to PCLS?
5. In Figure 4B the authors assessed serum antibodies but what mucosal IgA and IgG in BAL fluid? This may be more critical with a mucosally administered drug. Also do these antibodies neutralize the drug?
6. The results section is loaded with text that are methods and not results. For example lines 231-238 is text that dhow be in Methods or Figure Legends and not results per se.

Reviewer #2

(Remarks to the Author)

The authors previously reported the dimeric fusion lipopeptide and the in vivo efficacy study in ferrets. In this manuscript, they report a detailed study on the same peptide in K18-hACE2 mice. The study is quite thorough, looking at the safety, toxicity and efficacy comprehensively. They also looked at the transcriptome upon peptide treatment, which essentially reversed the changes caused by viral infection. The brain damage by cytokin, lipopeptide treatment decreased brain cytokine study explained the death of this model animal very well.

Overall, I think the study fits the journal pretty well, even though the study is still on the same peptide, but the information within the paper is quite valuable to the community.

Reviewer #3

(Remarks to the Author)

This article discusses the high inhibitory activity of monomer-PEG24 and dimer-PEG4 against SARS-CoV-2-S mediated fusion and virus infection in vitro and ex vivo. Both pre-treatment and post-treatment with fusion inhibitory peptides can effectively protect and treat mice, suggesting that lipopeptides may increase the survival rate of mice infected with SARS-CoV-2 by limiting the production of cytokines related to increased permeability of the blood-brain barrier, thereby reducing

the spread of virus into brain. From the transcriptomics perspective, it has been found that lipopeptides can reduce the inflammatory response in the lungs of mice infected with SARS-CoV-2. It is interesting to find that the secondary lipopeptide treatment still maintained high therapeutic activity even in the presence of peptide-specific antibodies. However, major revisions are needed to improve the quality of the current manuscript.

--Given that Omicron variants dominate the current COVID-19 epidemic, the efficacies of the inhibitors should be also evaluated in vitro and in vivo with the Omicron variants.

--The results of Figs 1A-D repeat the previous description lacking the novelty.

--Fig1C: which S protein was used in the cell-cell fusion assay? D614?

--Figs 1E-F: Considering the changes in the fusion pathways of diverse variants, Omicron variants can be used.

--Figs 2B-C: the monomer-PEG-24-treated mice show a 100% survival rate, but their body weight data are incomplete and different from the Dimer-PEG4 group mice. The reason should be given.

-- Transcriptomic data: the method used in Figure 3B, which involves summing the normalized counts per sample, has certain limitations. This approach does not take gene length differences into account when comparing gene expression across different genes. As a result, it may overestimate the expression levels of longer genes, as these naturally generate more reads in bulk RNA-seq data. This makes the method less suitable for comparisons between different genes. It is advisable to use normalization methods that account for gene length differences, such as TPM or RPKM/FPKM, when comparing different gene sets. This approach allows for a more accurate comparison of gene expression levels, reducing potential bias introduced by gene length differences. Additionally, performing Sample Level Enrichment Analysis (SLEA) using the normalized TPM or RPKM/FPKM values could help minimize biases related to gene length and sequencing depth differences, leading to more reliable results.

--Fig.4E legend: 6-10 dpi or 10 dpi?

--Fig.5D: statistically significant or not? While the brain VL are shown 2 dpi and 5dpi, the lung VL for 5 dpi should be shown.

--Lines 379-381: As reported, EK1-based inhibitors are derived from OC43 S rather than the SARS-CoV-2 S. Correct references should be cited here, for example with IPB02 and IPB24 inhibitors.

--Lines 433-437: There are two SARS-CoV-2 fusion inhibitory lipopeptides are currently evaluated in clinical phase II/III trials through nebulization or aerosols (Wu et al. *Sci China Life Sci* 2023; Zhu et al. *MedComm*, 2024). These should be discussed in this paragraph.

--Some Tyros, for example: Fig.2 (H); Line312 peptide-protected; the format of references, e.g., 5 and 7, 40, 71.

Version 1:

Reviewer comments:

Reviewer #1

(Remarks to the Author)

The authors have addressed my concerns.

Reviewer #2

(Remarks to the Author)

I think the authors have addressed all the comments and I recommend the publication of the paper.

Reviewer #3

(Remarks to the Author)

The authors have satisfactorily revised the manuscript, and this reviewer has no further questions.

Point-by-point responses to reviewers' comments

We thank reviewers for their insightful and constructive feedback. In response to raised questions, we completed the manuscript with the required information, performed additional experiments, reorganized the figures, adapted the size of the abstract and text to the journal guidelines and updated some references. We have addressed each comment and included bellow point-by-point list of the revision, and highlighted in yellow the text modified according to reviewers' suggestions. We hope that the revised manuscript, together with these detailed responses, resolves all the points raised by the reviewers.

Reviewer #1:

The paper by Mougari et al describe a lipopeptide formulation derived from HRC domain of S inhibit viral infection in vitro and show efficacy in both prophylaxis and post exposure prophylaxis in the K18ACE2 model of SARS-CV2 infection. Interestingly, the peptides do not interfere with convalescent anti-viral antibody response. Overall, the experiments are well performed, and the data support the author's conclusions. There are some points that need to be clarified.

Authors: We thank reviewer for the positive evaluation of our manuscript and instructive comments. We have made every effort to follow the recommendations and have revised the manuscript to address and clarify the raised points.

1. The 20 µL volume of peptide delivery will result in the material being aspirated into the lung as demonstrated by the authors. Thus the current dataset does not clarify whether an intranasal formulation or an inhaled formulation would be needed to move this program to the clinic. Do the authors have data with smaller volumes, such as 10 µL which have been shown to reduce lung delivery? Some of these points about formulation and route of delivery should also be added to the Discussion.

Authors:

The 20 µL volume was chosen based on previous studies in mouse models and determined to effectively deliver treatments without excessive lung aspiration, thus ensuring proper exposure to the nasal mucosa and respiratory. This volume is often used for vaccine administration and respiratory pathogen inoculation in mice (examples: PMID: 38714446, PMID: 35352010). We have used the smaller volume, such as 10 µl, in previous studies where measles virus specific peptides were administrated intranasally, resulting with the successful protection from viral infection (PMID: 27733647). In the murine model of intranasal infection we tend to administrate the peptide in the similar volume as a virus, to assure that the coverage of the respiratory tract corresponds to the similar surface that which will come in contact with the virus. The formulation that would be needed to move this approach to the clinic requires additional studies, which has been discussed in the last paragraph of the discussion in the revised manuscript.

2. The authors perform repeated dosing to assess toxicity but more relevant would be to assess olfaction in mice (<https://pubmed.ncbi.nlm.nih.gov/31641151/>) to make sure the peptides are not toxic to olfactory neurons.

Authors:

Lipopeptides used in our study contain 36 amino-acids from the HRC domain of SARS-CoV-2 Spike protein, conjugated to Polyethylene glycol (PEG) and Cholesterol molecules. PEG and cholesterol are found in the composition of different types of therapeutic molecules delivered to lungs via intranasal routes (PMID:17244708, PMID:24955820) and their toxicity to olfactory neurons has not been reported so far neither in mice nor in humans. In addition, different constructions of viral vectors inducing the expression of the Spike and administrated via the intranasal route have been tested as vaccines against SARS-CoV-2 infection, but their toxicity has never been reported (PMID: 32931734, PMID: 33357418). Concerning study of Valli et al. indicated by the reviewer, the investigation of toxicity on the olfactory function is certainly more warrant than in our study as the LTA1 used in that study is derivative of the heat-labile toxin *E. coli* (LT), which is known to be responsible for olfactory dysfunctions.

Nevertheless, in response to the reviewer comment, we tested the toxicity of both peptides, monomer-PEG24 and dimer-PEG4, in organotypic cultures prepared from murine olfactory bulbs, cerebellum and lungs by daily treatment of the cultures with lipopeptides for 3 days. The slices were then analyzed using MTT test (3-(4,5-dimethylthiazol-2-yl)-2,5-diphenyltetrazolium bromide) to assess potential cytotoxic effect of peptides following the treatment and the test is described in the method section of the revised manuscript. We observed that the metabolic activity in treated slices was similar to controls treated with medium in all analyzed organotypic cultures, thus reinforcing the aspect of the safety of lipopeptide treatment. These results are added to the supplementary material (new Fig. S2) and discussed in the revised version of the manuscript (lines 181-185).

3. In Figure 2, why is there no effect of the peptide on CNS viral load at day 2?

Authors:

As SARS-CoV-2 is a respiratory virus, the peptides were administrated by respiratory route to be able to act at the place of viral entry, and decrease the transfer of the virus in the brain in the later stages of infection. Indeed, lipopeptides induced the significant reduction of viral load in lungs as well as in oral swabs, and viral load was reduced in the brain at day 5, although not significantly, most probably due to the small number of samples obtained for analysis. It is highly probable that although lipopeptides could reduce virus replication in lungs at the early stage of infection, the virus is still able to spread to the CNS later after infection, probably via the olfactory bulb as demonstrated in Hamster model (PMID: 37495586). This may explain the absence of effect on viral load in CNS at day 2. We hypothesize that due to the lower viral load in peptide-treated animals, their immune system is able to develop an adequate response to control the infection in CNS in later stages, which is followed with the much higher survival rate of treated animals. In the case of nontreated animals, we proposed that the proinflammatory cytokines, upregulated in absence of treatment, induced a second entry route into the brain via the permeabilization of the brain-blood barrier, thus resulting in a massive entry of virus into CNS via the blood as attested by the high virus titer observed in the brains of nontreated mice at day 6 p.i. This has been discussed now in the revised version of the manuscript (lines 419-420).

4. For the organotypic cultures, were the lungs inflated with LMP agarose similar to PCLS?

Authors:

The organotypic lung cultures used in the manuscript present a different model where the lung slices are kept in their native forms in culture, and do not require the utilization of a low melting agarose to inflate the lungs. Our protocol gave good results in terms of viability and response to infection and was described in detail in a previous paper from the group (PMID: 34608167).

5. In Figure 4B the authors assessed serum antibodies but what mucosal IgA and IgG in BAL fluid? This may be more critical with a mucosally administered drug. Also do these antibodies neutralize the drug?

Authors:

We agree with the reviewer that quantifying mucosal antibodies could be important for the evaluation of the possible anti-peptide response in peptide-treated mice, in addition to their evaluation in serum. We could not exclude that they are generated in some mice, similarly to what we have observed with serum IgG. Nevertheless, the major aim of that experiment was to test if the repeated administration of the high dose peptides, which may be immunogenic in mice, could interfere with the antiviral activity. We believe that results presented in the Figure 4 fully answered to that question, presenting the successful antiviral protection of peptides. The point raised by the reviewer was developed in the discussion of the revised manuscript (lines 429-430).

6. The results section is loaded with text that are methods and not results. For example lines 231-238 is text that should be in Methods or Figure Legends and not results per se.

Authors:

The paragraph describing the protocol has been added to help the visualization of the experimental design of the study, critical for understanding the results presented in the following paragraph as this format of presenting results alongside relevant experimental details is commonly found in articles published in this journal. In response to reviewer's comment, the sentence describing the method without any reference to results has been moved to the Methods section, in Transcriptomic analysis chapter (lines 641-650)

Reviewer #2 (Remarks to the Author):

The authors previously reported the dimeric fusion lipopeptide and the in vivo efficacy study in ferrets. In this manuscript, they report a detailed study on the same peptide in K18-hACE2 mice. The study is quite thorough, looking at the safety, toxicity and efficacy comprehensively. They also looked at the transcriptome upon peptide treatment, which essentially reversed the changes caused by viral infection. The brain damage by cytokines, lipopeptide treatment decreased brain cytokine study explained the death of this model animal very well.

Overall, I think the study fits the journal pretty well, even though the study is still on the same peptide, but the information within the paper is quite valuable to the community.

Authors:

We thank reviewer for the positive evaluation of our manuscript.

Reviewer #3 (Remarks to the Author):

This article discusses the high inhibitory activity of monomer-PEG24 and dimer-PEG4 against SARS-CoV-2-S mediated fusion and virus infection in vitro and ex vivo. Both pre-treatment and post-treatment with fusion inhibitory peptides can effectively protect and treat mice, suggesting that lipopeptides may increase the survival rate of mice infected with SARS-CoV-2 by limiting the production of cytokines related to increased permeability of the blood-brain

barrier, thereby reducing the spread of virus into brain. From the transcriptomics perspective, it has been found that lipopeptides can reduce the inflammatory response in the lungs of mice infected with SARS-CoV-2. It is interesting to find that the secondary lipopeptide treatment still maintained high therapeutic activity even in the presence of peptide-specific antibodies. However, major revisions are needed to improve the quality of the current manuscript.

Authors:

We thank reviewer for the constructive review of our manuscript. We have carefully considered all recommendations and made the necessary revisions to improve the manuscript and the presentation of the results.

--Given that Omicron variants dominate the current COVID-19 epidemic, the efficacies of the inhibitors should be also evaluated *in vitro* and *in vivo* with the Omicron variants.

Authors:

The effect of lipopeptides has been evaluated against Omicron variant *in vitro* in our previous studies (PMID: 35695453). In this study, we initially tested the susceptibility of K18-hACE2 mice to different variants of SARS-CoV-2 (Fig. S4) and observed that Omicron variant does not cause serious disease in infected mice. We have also quantified the viral load in different organs of Omicron-infected mice at the time of euthanasia (day 11 or 18 p.i.) and observed a consistently lower viral load using qRT-PCR in all organs compared to the other variants like the alpha (new Fig. S4F). We therefore concluded that the peptide efficacy could not be correctly evaluated against Omicron in the K18-hACE2 mouse model as this variant causes only limited infection with rapid resolution. This is now additionally commented in the revised version (lines 196-197).

It is also noteworthy to state that the dominant SARS-CoV-2 variants that are currently circulating are derivative from BA.2 variant and have been also shown to cause only low disease manifestations in K18-hACE2 mice (PMID: 36084644, PMID: 39018495). The Wuhan, Alpha and Delta variants were selected for this study due to their high pathogenicity in hACE2 mice, with the aim of testing the peptide's efficacy using reliable *in vivo* biomarkers, including reducing infection, preventing lethality, and improving the clinical manifestations of infection. On the other hand, in a previous study, we have demonstrated the capacity of our peptides to block the process of membrane fusion (fusion assay) and virus entry (live virus assay) against different variants of SARS-CoV-2 including BA.1 and BA.2 (the ancestral strain of the dominant variants currently circulating), allowing to validate the antiviral activity of peptide against the Omicron derivative variants (PMID: 35695453). Moreover, we have also previously demonstrated that peptide efficacy extends to block the process of membrane fusion against other coronaviruses such as SARS-CoV-1 and Middle East respiratory syndrome coronavirus (MERS-CoV) and also block infection by live MERS-CoV in Vero cells (PMID: 33082259). Altogether, these results validate the broad-spectrum efficacy of the strategy based on targeting the conserved HRC-domains to block fusion of SARS-CoV-2 variants.

--The results of Figs 1A-D repeat the previous description lacking the novelty.

Authors:

We believe that Figs. A-B provide valuable insights for readers, as they illustrate the peptide structures used throughout this study and facilitate further reading. Figs. C-D present data obtained with a new peptide batch in a different laboratory, to confirm the reproducibility of our approach, as mentioned in line 156 of the revised manuscript. In addition, we believe that it is important to present those results to introduce the further studies performed in the manuscript and allow the comparison between *in vitro* and *ex vivo* data obtained with the same

lipopeptides. Finally, figs. 1E-G present novel data on peptide efficacy in lung organotypic cultures.

--Fig1C: which S protein was used in the cell-cell fusion assay? D614?

Authors:

The S protein used in the fusion test was the D614. This was now pointed out in the revised version of the figure 1 legend.

--Figs 1E-F: Considering the changes in the fusion pathways of diverse variants, Omicron variants can be used.

Authors:

The fusion pathway of SARS-CoV-2 variants has indeed evolved, with Omicron variants suggested to preferentially enter host cells through the endo-lysosomal pathway-mediated membrane fusion, rather than the TMPRSS2-driven plasma membrane fusion observed with earlier variants like the Wuhan and Delta strain. While this point has been debated in recent studies conducted using relevant *ex vivo* models (PMID: 37555660), and as stated in our response to the first reviewer's question, our previous study established that lipopeptides used in the manuscript retain their antiviral activity against this type of fusion pathways. The effect of lipopeptides has been evaluated against Omicron variant and the Omicron derivative BA.1 and BA.2 *in vitro* in one of our previous studies (PMID: 35695453). This point has now been discussed in the revised version of the manuscript (lines 390-393).

--Figs 2B-C: the monomer-PEG-24-treated mice show a 100% survival rate, but their body weight data are incomplete and different from the Dimer-PEG4 group mice. The reason should be given.

Authors:

Both treated groups were monitored for their survival for at least 21 days post-infection (dpi). Additional time points have been added in the revised version of the Fig 2B.

-- Transcriptomic data: the method used in Figure 3B, which involves summing the normalized counts per sample, has certain limitations. This approach does not take gene length differences into account when comparing gene expression across different genes. As a result, it may overestimate the expression levels of longer genes, as these naturally generate more reads in bulk RNA-seq data. This makes the method less suitable for comparisons between different genes. It is advisable to use normalization methods that account for gene length differences, such as TPM or RPKM/FPKM, when comparing different gene sets. This approach allows for a more accurate comparison of gene expression levels, reducing potential bias introduced by gene length differences. Additionally, performing Sample Level Enrichment Analysis (SLEA) using the normalized TPM or RPKM/FPKM values could help minimize biases related to gene length and sequencing depth differences, leading to more reliable results.

Authors:

We thank reviewer for the suggestion, which indeed allows a more accurate comparison of gene expression levels. We performed the normalization of the transcriptomic data, using sum of transcripts per million (TPM) per sample to control for the effect of transcript length, and determine expression sums per gene category. The obtained results, comparing different gene sets are now presented in the new figure 3B and the approach is mentioned additionally in the Methods section (lines 640-650) and the figure legend. This approach revealed difference between analyzed groups in the first two presented clusters to be even more statistically significant.

--Fig.4E legend: 6-10 dpi or 10 dpi?

Authors:

It is 6-10 dpi. This has been now modified the legend of the Fig.4E.

--Fig.5D: statistically significant or not? While the brain VL are shown 2 dpi and 5dpi, the lung VL for 5 dpi should be shown.

Authors:

This figure was completed with lung viral load at 5 dpi where the differences were not significant and the results are commented in the revised version of the manuscript (lines 321-322).

--Lines 379-381: As reported, EK1-based inhibitors are derived from OC43 S rather than the SARS-CoV-2 S. Correct references should be cited here, for example with IPB02 and IPB24 inhibitors.

Authors:

As suggested, this has been corrected in the revised version of the manuscript and the corresponding reference, N°62 was added (line 388).

--Lines 433-437: There are two SARS-CoV-2 fusion inhibitory lipopeptides are currently evaluated in clinical phase II/III trials through nebulization or aerosols (Wu et al. Sci China Life Sci 2023; Zhu et al. MedComm, 2024). These should be discussed in this paragraph.

Authors:

These studies have been added (references N° 70 and 71) and have been now discussed in the last paragraph of the discussion of the revised manuscript (lines 447-448).

--Some Tyros, for example: Fig.2 (H); Line312 peptide-protected

Authors:

We thank reviewer for pointed out these omissions, which have been corrected in the revised version of the manuscript.

--The format of references, e.g., 5 and 7, 40, 71.

Authors:

The references have been updated to align with the version indicated by the publishing journal, while respecting the format of this journal.